# Rethinking Radiology Report Generation: From Narrative Flow to Topic-Guided Findings

**Sheng Cheng & Devika Subramanian**[*]
Department of Computer Science
Rice University
6100 Main St., Houston, Texas, 77005, USA
{sc159,devika}@rice.edu

## Abstract

Vision-Language Models (VLMs) for radiology report generation are typically trained to mimic the narrative flow of human experts. However, we identify a potential limitation in this conventional paradigm. We hypothesize that optimizing for narrative coherence encourages models to rely on linguistic priors and inter-sentence correlations, which can weaken their grounding in direct visual evidence and lead to factual inaccuracies. To investigate this, we design a controlled experiment demonstrating that as textual context increases, a model's reliance on the input image systematically decays. We propose LLaVA-TA (Topic-guided and Anatomy-aware), a new fine-tuning framework that directly addresses this challenge by re-engineering the generation process. Instead of producing a linear narrative, LLaVA-TA decomposes the report into a set of independent, clinically-relevant topics. By training the model to generate a discrete finding for each topic conditioned on both the full image and its corresponding anatomical region, we reduce the model's reliance on narrative flow and enforce stricter visual grounding. Our experiments show that LLaVA-TA sets a new state of the art on the MIMIC-CXR dataset, significantly improving clinical accuracy on metrics like RadGraph F1 (from 29.4 to 44.0) and CheXpert F1-14 (from 39.5 to 71.5) over strong baselines. Our work demonstrates that dismantling a report's narrative structure to enforce independent, visually-grounded observations is a crucial and effective step toward building more accurate and reliable medical VLMs.

## 1 Introduction

The automated generation of radiology reports from medical images, such as chest X-rays (CXRs), is a pivotal task in medical AI with the potential to streamline clinical workflows and reduce radiologist burnout. Unlike general-purpose vision-language tasks, radiology report generation (RRG) (Chen et al., 2022; Tanida et al., 2023; Tanno et al., 2025) operates with minimal tolerance for error, demanding exceptional factual precision, spatial grounding of findings to specific anatomy, and adherence to structured medical language—where ambiguity can have severe clinical consequences.

We propose that a potential pitfall in adapting large Vision-Language Models (VLMs) (Wang et al., 2024; Liu et al., 2023; Grattafiori et al., 2024; OpenAI et al., 2024) to RRG lies in their implicit optimization for narrative flow. Our central hypothesis is that conventional training, by treating the report as a monolithic text sequence, inadvertently teaches models to prioritize linguistic coherence over strict visual fidelity. This learned reliance on the textual context of previously generated sentences could become actively detrimental, allowing it to bias subsequent findings and override direct evidence from the image itself—a process that we suspect weakens visual grounding and fosters clinical hallucination. To investigate this vulnerability, we designed a controlled experiment with a pre-trained VLM, LLaVA-Rad (Zambrano Chaves et al., 2025). We prompted the model to complete the last K sentences of a ground-truth report given the preceding sentences, comparing its

---

[*]Corresponding author.

performance when conditioned on the true CXR versus a blank image. Our findings provide strong empirical validation for our initial hypothesis, as shown in Fig. 1. Crucially, the performance gap between using the real versus the blank image diminished significantly as the textual prefix grew longer (i.e., as K decreased). This demonstrates that with sufficient textual context, the model increasingly disregards the visual input, relying instead on learned language patterns to complete the report—a critical vulnerability for a diagnostic task.

This core issue of narrative-induced bias persists across existing methodologies. For instance, recent works like MAIRA-2 (Bannur et al., 2024) and DART (Park et al., 2025) have introduced explicit grounding objectives and self-correction modules to better align text with disease regions. Similarly, approaches such as Multi-Phased Supervision (Chen et al., 2025) and Context-Enhanced RRG (Delbrouck et al., 2025) attempt to enforce clinical accuracy through hierarchical training or rich context integration. However, despite these advancements in alignment, these models largely retain the autoregressive architecture that allows earlier generated tokens to dictate the probability of future findings, leaving the narrative bias intact. Even specialized medical VLMs have not fully escaped this trap (Li et al., 2023; Chen et al., 2024; Hyland et al., 2024; Tu et al., 2023). LLaVA-Rad, for instance, improves performance by filtering out sentences describing historical comparisons but still generates the core findings as a sequential narrative. Other notable approaches, such as COMG (Gu et al., 2024), which incorporates anatomical masks, or the Multi-Grained framework (Liu et al., 2024) with its sentence-level contrastive learning, have advanced the field. Yet, they do not fundamentally dismantle the sentence-to-sentence dependency that allows contextual priors to override direct visual evidence, failing to achieve topic-level disentanglement.

To address this fundamental limitation, we propose LLaVA-TA (Topic-guided and Anatomy-aware), a novel fine-tuning framework that re-envisions report generation. Instead of producing a linear narrative, LLaVA-TA decomposes the task by distinct medical topics corresponding to anatomical regions (e.g., lungs, heart, mediastinum). The model is trained to generate short, independent paragraphs for each relevant topic, conditioned on the entire image. This topic-guided approach confers two critical advantages. **First, it enhances visual grounding.** By breaking the sequential chain of dependency, we force the model to ground its statements for each topic (e.g., "lungs") directly and solely on the visual evidence present in the corresponding anatomical area, preventing knowledge from a previously generated sentence (e.g., about the "heart") from biasing the text. The visual grounding is provided by a CXR segmentation model that provides masks for each topic, as shown in Fig. 2. **Second, it reduces hallucination via tighter supervision.** Training on short, topic-isolated ground-truth sentences provides a much stricter and more focused learning signal. This structure minimizes the model's ability to make "lucky guesses" based on spurious sentence correlations, a common failure mode during training that reinforces hallucinatory behavior.

Our key contributions are as follows:

- **Diagnosing the "Narrative Flow" Problem.** We empirically validate that narrative-flow-based training is a fundamental limitation in RRG, demonstrating how it induces a harmful trade-off between textual coherence and visual grounding.
- **LLaVA-TA: A Topic-Guided Generation Framework.** We propose LLaVA-TA, a novel framework that re-engineers the generation process by decomposing it into independent, clinically relevant topics to enforce stricter visual grounding.
- **State-of-the-Art Clinical Accuracy and Grounding.** We demonstrate through extensive experiments that LLaVA-TA sets a new state of the art, producing reports that are significantly more factually accurate, spatially grounded, and less prone to clinical hallucination than prior methods.

## 2 METHODS

The LLaVA-TA framework enhances RRG by replacing unstructured, narrative-based training with structured supervision that explicitly aligns clinical topics with their corresponding anatomical regions in the chest X-ray. We first define "Narrative Bias" as follows: Narrative Bias is the tendency of an autoregressive model $P(y_t|y_{<t}, I)$ to maximize the likelihood of the next token $y_t$ based primarily on the linguistic probability $P(y_t|y_{<t})$ rather than the visual conditional probability $P(y_t|I)$. As illustrated in Fig. 2, LLaVA-TA comprises three core stages to address this issue: (1) topic-guided

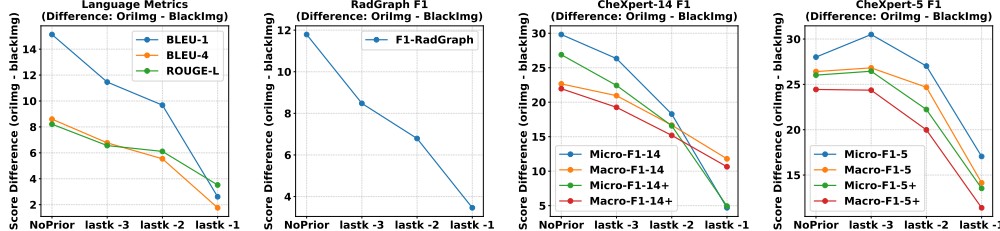

Figure 1: Empirical validation of the model's over-reliance on textual priors over visual evidence. The plots show the score difference between generating report findings with the original image versus a blank image across four metric categories. The y-axis represents this performance gap (OriImg - BlackImg); a larger value means the visual information provides a greater benefit. The x-axis represents the amount of textual context provided to the model, where "NoPrior" is full generation and "lastk -K" is the task of generating only the final K sentences. Across all metrics, the performance gap consistently shrinks as more textual context is given (i.e., as K decreases from 3 to 1). This demonstrates that the model increasingly ignores the visual input as the textual prefix grows, relying instead on learned linguistic patterns to complete the report. This result provides strong evidence that training with narrative flow weakens visual grounding. Detailed results are shown in Appendix A.

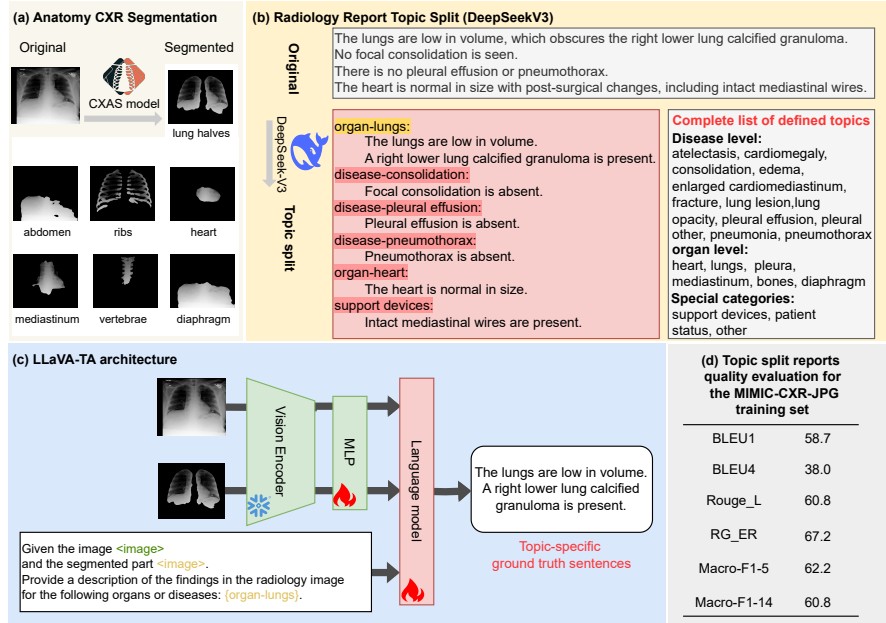

Figure 2: LLaVA-TA framework. Our pipeline transforms unstructured data into aligned, topic-specific training instances. (a) First, a pretrained segmentation model (CXAS) generates anatomy-specific masks for each CXR to serve as spatially-aware visual inputs. (b) Concurrently, an LLM (DeepSeek-V3) dismantles the narrative structure of each report, decomposing it into discrete, cleaned sentences aligned with a clinical topic ontology. (c) During fine-tuning, the LLaVA-TA model is conditioned on both the original image and the relevant anatomical mask, and is prompted to generate a finding for a single topic. (d) A quantitative evaluation confirms that our report decomposition process preserves high lexical and clinical fidelity compared to the original reports.

decomposition of ground-truth reports into atomic, image-grounded statements; (2) anatomical segmentation to create spatially-aware visual inputs; and (3) a vision-language fine-tuning process that learns to generate topic-specific findings from these aligned pairs.

## 2.1 TOPIC-GUIDED REPORT DECOMPOSITION

Our first step is to disentangle the complex, multi-topic narratives of clinical reports into clean, image-grounded training targets. To achieve this, we employ DeepSeek-V3 (DeepSeek-AI et al., 2025), an instruction-following LLM, to preprocess and decompose each report into a set of discrete, topic-specific sentences. The topics are shown in Fig. 2(b).

This process consists of two key steps:

- **Cleaning:** Comparative phrases (e.g., "compared to prior exam") and speculative language (e.g., "possibly atelectasis") are removed to retain only evidence-based findings visible in the current image.

- **Topic Splitting:** Each sentence is mapped to a clinical topic using a hierarchical ontology. Topics are prioritized as follows: (1) pathology-level (e.g., "disease-consolidation"), (2) anatomy-level (e.g., "organ-lungs"), and (3) auxiliary categories (e.g., "support devices").

To ensure high-quality sentence splits, we apply additional rules: each sentence must refer to a single topic, comparative or ambiguous language is rewritten for clarity, and negations are standardized (e.g., "Pneumothorax is absent"). This yields structured, direct, and image-grounded sentence targets for each report. The complete list of topics are provided in the Appendix B. The full prompt for applying the topic-split is shown in the Appendix C.

Each radiology report is transformed into a set of clean, topic-specific findings. This process enforces one sentence per topic and provides clear supervision targets that can be aligned with localized visual inputs.

We further evaluate the quality of the topic-split reports using both lexical metrics and radiology-specific metrics over the entire MIMIC-CXR-JPG training set (Johnson et al., 2019b) by concatenating them as a whole paragraph first; the results are shown in Fig. 2(d). The evaluation demonstrates strong alignment between preprocessed and original medical reports, with high BLEU-1 (58.7), BLEU-4 (38.0) and ROUGE-L (60.8) scores confirming lexical and semantic fidelity. The superior RadGraph-F1 (67.2) highlights robust retention of structured clinical entities and relations, as this metric extracts clinical entities (e.g., "lung", "opacity") and the relations connecting them (e.g., "opacity" $\rightarrow$ "located in" $\rightarrow$ "lung"), computing the F1 overlap of these semantic triplets, while Micro-F1-14 (0.64) and Macro-F1-14 (0.61) indicate effective fine-grained label preservation.

## 2.2 ANATOMICAL REGION SEGMENTATION

To provide spatial grounding for each textual topic, we generate anatomical segmentation masks that isolate the relevant visual regions within the CXR. We employ CXAS (Seibold et al., 2022; 2023), a specialized UNet-based model, to segment each image into seven key anatomical regions: *vertebrae, ribs, diaphragm, mediastinum, abdomen, heart*, and *lung halves*.

Each clinical topic from our ontology is mapped to one or more of these anatomical regions via a predefined lookup table (Appendix B). If a topic pertains to multiple regions (e.g., "disease-pleural effusion" may involve the lungs and diaphragm), their corresponding masks are merged. This mapping yields two aligned visual inputs for each topic-sentence pair: (1) the original global image $I$, and (2) the topic-specific segmented image $S$, which provides a localized view. This dual-input strategy focuses the model's attention, improving interpretability and reducing noise from irrelevant regions.

## 2.3 VISION-LANGUAGE FINE-TUNING

Our fine-tuning process, adapted from LLaVA-Rad, trains a model to generate topic-specific findings conditioned on both global and localized visual evidence. The architecture consists of three core components: (1) **Vision Encoder:** BiomedCLIP-CXR (Zhang et al., 2025; Zambrano Chaves et al.,

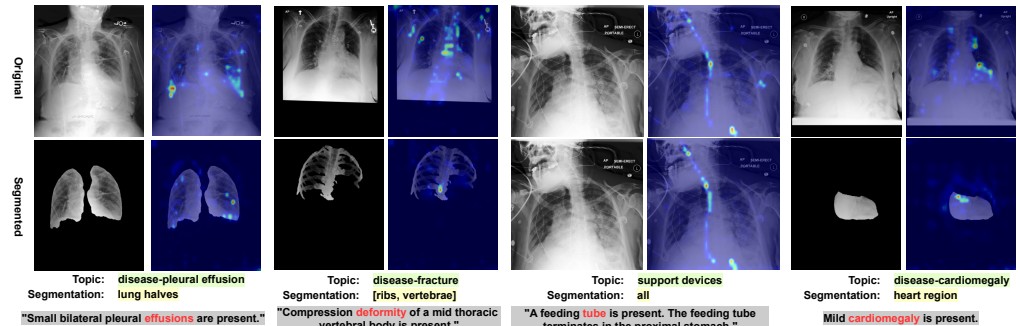

Figure 3: Attention Visualization of LLaVA-TA. We adopt an aggregation using max across all attention layers and heads of the attention scores for the currently generated tokens for the features corresponding to the two input images. The examples above illustrate how LLaVA-TA grounds clinically meaningful tokens in the appropriate image regions, demonstrating spatial interpretability for specific findings in the generated report (bottom row).

2025), pretrained on 697k paired radiology image-report data, processes both the original image $I$ and the segmented image $S$, producing two distinct visual embeddings, $Z_I$ and $Z_S$. (2) **Multimodal Alignment Layer:** A learnable MLP projects the visual embeddings $Z_I$ and $Z_S$ from the vision encoder's feature space into the language model's word embedding space. (3) **Language Model:** Vicuna-7B-v1.5 (Zheng et al., 2023), an instruction-tuned LLM, is used to generate topic-specific findings auto-regressively.

Each training instance is constructed from a structured prompt containing two special <image> tokens: *"Given the image <image> and the segmented part <image>, describe the findings for the topic: topic."*. During the forward pass, the token embeddings corresponding to the two <image> placeholders are replaced by the projected visual embeddings from the MLP. This creates a single, unified input sequence of embeddings that seamlessly integrates textual and visual information for the language model.

The model is trained to maximize the likelihood of the ground-truth topic sentence $Y = (y_1, ..., y_L)$ by minimizing the standard auto-regressive cross-entropy loss:

$$\mathcal{L}(\theta) = -\sum_{t=1}^{L} \log P(y_t | y_{<t}, X_{\text{prompt}}, Z_I, Z_S; \theta)$$

where $\theta$ represents the trainable model parameters, $y_t$ is the $t$-th token in the target sentence, $y_{<t}$ are the preceding ground-truth tokens, and $X_{\text{prompt}}$ is the structured text prompt.

We follow the established two-phase training protocol: (1)**Phase 1: Cross-Modal Alignment.** The vision encoder and LLM are kept frozen. Only the MLP is trained to align the visual features with the LLM's token space. (2)**Phase 2: Instruction Fine-Tuning.** The vision encoder remains frozen while the MLP and the full LLM are fine-tuned end-to-end using the auto-regressive loss objective defined above. This phase teaches the model to generate coherent, topic-specific sentences that are conditioned on both the global and localized image features.

## 2.4 INFERENCE PROCESS

During inference, we evaluate our model under two distinct settings to provide a comprehensive assessment. The first setting measures fine-grained, topic-level accuracy, while the second evaluates the model's ability to generate a complete and clinically coherent report.

### 2.4.1 TOPIC-LEVEL EVALUATION

This setting provides a controlled evaluation of the model's core ability to generate a factually correct sentence for a given clinical topic. During evaluation, each test instance consists of the image pair $(I, S)$ and one ground-truth topic (e.g., "disease-consolidation") as the prompt. The single sentence generated by the model is then compared directly against the corresponding ground-truth sentence

for that specific topic. This process isolates the model's conditional text generation quality from the separate challenge of topic selection.

### 2.4.2 REPORT-LEVEL EVALUATION

This setting assesses the model's practical ability to synthesize a complete and comprehensive clinical report. To achieve this, the model iterates through a set of topics and generates a sentence for each; the final report is the concatenation of these individual outputs. The topic set is compiled as follows: (1) **Organ- and special- level topics.** To ensure systematic and comprehensive anatomical coverage, the model is prompted with the complete, predefined set of all organ-level and auxiliary topics for every report. (2) **Disease-level topics.** We use the ground-truth disease labels from the test set as prompts. This is a deliberate experimental design choice to isolate the generative model's performance from the variance and potential errors of an upstream disease classification model, enabling a fair and direct comparison of text generation quality against other methods. The complete, multi-sentence generated report is then evaluated against the full ground-truth report.

## 3 IMPLEMENTATION DETAILS

### 3.1 DATASET

We conduct training and evaluation on the MIMIC-CXR-JPG dataset (Johnson et al., 2019a;b), following the official train/validation/test splits. We retain only frontal-view images and exclude studies that lack a "Findings" section, consistent with the filtering protocol of LLaVA-Rad. After preprocessing, we obtain 212,379 training, 1,721 validation, and 3,029 test image-report pairs.

Following our topic-splitting procedure using DeepSeek-V3, each report is decomposed into multiple topic-level sentences. This results in 1,161,753 topic-aligned training pairs, 9,388 validation pairs, and 17,193 test pairs—each consisting of an image, a localized anatomical region, and a list of topic-specific sentences.

To evaluate cross-domain generalization, we apply the same preprocessing pipeline to the IU-Xray dataset (Demner-Fushman et al., 2015), yielding 2,950 images and 16,131 image-topic pairs after topic decomposition and filtering.

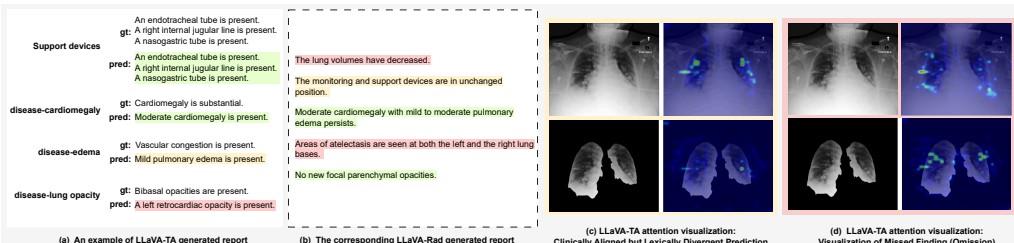

Figure 4: Qualitative example of LLaVA-TA's report generation and attention visualization. (a) shows the report generated by LLaVA-TA; (b) shows the output from the baseline model LLaVA-Rad. Green highlights denote correct predictions, orange indicates partially correct sentences (including hallucinations), and red marks omissions. (c) and (d) visualize LLaVA-TA's attention for the partially correct and omitted findings, respectively. Notably, even when LLaVA-TA fails to fully verbalize a finding, its attention maps correctly highlight the lesion area in the image. This spatial grounding provides interpretable feedback, helping radiologists identify potential errors, thereby increasing trust in model outputs.

### 3.2 TRAINING DETAILS

We adopt a two-stage training procedure and the same hyper-parameters as LLaVA-Rad. In stage 1, we first train the multimodal alignment MLP while keeping the vision encoder and language model frozen. This stage runs for 1 epoch on the MIMIC-CXR-JPG training set. We use a batch size of 256 and a learning rate of $1e^{-3}$, with a cosine scheduler and warm-up ratio of 0.03. In stage 2, we fine-tune both the language model and MLP using LoRA (Hu et al., 2021) with a rank of 128.

The vision encoder remains frozen. Training is performed for 3 epochs on the MIMIC-CXR-JPG training set, using a batch size of 64 and a cosine learning rate of $1\mathrm{e}^{-4}$ with a 0.03 warm-up ratio. We report results using the final model.

All experiments are conducted on 4 A100 GPUs (40GB), and our implementation builds on the LLaVA-Rad codebase with modifications to support topic-level supervision and dual-image input.

## 4 RESULTS

We evaluate LLaVA-TA to answer three key questions: (1) Does topic-guided generation outperform state-of-the-art narrative-based methods in clinical accuracy? (2) What are the specific contributions of topic disentanglement versus anatomy-aware grounding? (3) Does our approach generalize to new datasets and provide interpretable, visually-grounded outputs?

Table 1: Main performance on the MIMIC-CXR-JPG test set. LLaVA-TA establishes a new state of the art, significantly outperforming all baselines. The strong performance of our ablation, LLaVA-T, confirms that topic disentanglement is the primary driver of improvement. While the additional anatomical masks in LLaVA-TA offer comparable performance here, they are crucial for enabling parameter-efficient fine-tuning (Tables 2 and 4).

| Model | Size | CheXpert | | | | | | | | RadGraph F1 | BLEU | | ROUGE-L |
| | | Micro-avg | | Macro-avg | | Micro-avg | | Macro-avg | | | (1) | (4) | |
| | | F1-14 | F1-5 | F1-14 | F1-5 | F1-14+ | F1-5+ | F1-14+ | F1-5+ | | | | |
| R2Gen | ≤1B | - | - | - | - | 22.8 | 34.6 | - | - | - | 35.3 | 8.6 | 27.7 |
| RGRG | ≤1B | - | - | - | - | - | 54.7 | - | - | - | 37.3 | 12.6 | 26.4 |
| Flamingo-CXR | ≤1B | - | - | - | - | 51.9 | 56.5 | - | - | 20.5 | - | 10.1 | 29.7 |
| Qwen2-VL | 7B | 25.9 | 20.6 | 17.0 | 16.5 | 25.0 | 22.4 | 19.3 | 20.3 | 11.0 | 12.0 | 0.8 | 11.6 |
| LLaVA | 7B | 22.9 | 23.4 | 15.4 | 17.5 | 23.7 | 26.9 | 17.0 | 20.3 | 4.5 | 21.0 | 1.3 | 13.8 |
| LLaVA-Med | 7B | 27.2 | 22.0 | 15.5 | 16.6 | 27.3 | 24.4 | 18.7 | 20.5 | 6.5 | 22.2 | 1.0 | 13.3 |
| CheXagent | 7B | 39.3 | 41.2 | 24.7 | 34.5 | 39.4 | 42.1 | 27.3 | 35.8 | 20.5 | 16.9 | 4.7 | 21.5 |
| MAIRA-1 | 7B | 55.7 | 56.0 | 38.6 | 47.7 | 55.3 | 58.8 | 42.3 | 51.7 | 29.6 | 39.2 | 14.2 | 28.9 |
| GPT-4V | - | 35.5 | 25.8 | 20.4 | 19.6 | 35.6 | 33.3 | 25.3 | 29.6 | 13.2 | 16.4 | 1.9 | 13.2 |
| Llama 3.2 | 11B | 33.3 | 31.8 | 21.3 | 26.3 | 34.4 | 37.4 | 25.0 | 32.6 | 9.0 | 10.2 | 0.7 | 11.1 |
| Med-PaLM M | 84B | 53.6 | 57.9 | 39.8 | 51.6 | - | - | - | - | 26.7 | 32.3 | 11.3 | 27.3 |
| LLaVA-Rad | 7B | 58.1 | 57.7 | 40.1 | 50.3 | 57.7 | 59.8 | 42.9 | 53.0 | 30.8 | 40.0 | 16.1 | 30.8 |
| LLaVA-T (topic)* | 7B | **83.7** | **90.0** | **72.2** | **87.9** | **83.7** | **90.3** | **78.2** | **88.7** | **46.5** | **50.9** | **32.1** | **60.9** |
| LLaVA-T (report)* | 7B | 80.2 | 88.5 | 63.0 | 87.0 | 81.6 | 88.7 | 71.4 | 87.7 | 34.2 | 43.9 | 25.7 | 41.8 |
| LLaVA-TA (topic)* | 7B | 83.5 | 89.6 | 71.5 | 87.3 | 83.6 | 90.1 | 77.9 | 88.3 | 44.0 | 50.9 | 31.8 | 60.6 |
| LLaVA-TA (report)* | 7B | 80.0 | 89.0 | 62.4 | 86.8 | 81.0 | 88.7 | 70.7 | 87.4 | 34.3 | 43.7 | 24.8 | 42.6 |

Table 2: In-distribution performance on MIMIC-CXR with MLP-only fine-tuning. In this parameter-efficient setting, LLaVA-TA dramatically outperforms both the LLaVA-Rad baseline and our topic-only LLaVA-T ablation. The results demonstrate that when the LLM is frozen, the explicit spatial guidance from anatomical masks is critical for learning an effective mapping from vision to language.

| Model | Size | CheXpert | | | | | | | | RadGraph F1 | BLEU | | ROUGE-L |
| | | Micro-avg | | Macro-avg | | Micro-avg | | Macro-avg | | | (1) | (4) | |
| | | F1-14 | F1-5 | F1-14 | F1-5 | F1-14+ | F1-5+ | F1-14+ | F1-5+ | | | | |
| LLaVA-Rad | 7B | 42.1 | 48.5 | 23.3 | 36.6 | 40.6 | 48.8 | 25.1 | 39.0 | 19.8 | 16.5 | 3.6 | 15.3 |
| LLaVA-T (topic)* | 7B | 27.7 | 27.5 | 20.4 | 22.3 | 27.7 | 27.5 | 21.9 | 23.0 | 5.8 | 2.9 | 0.4 | 5.9 |
| LLaVA-T (report)* | 7B | 11.1 | 7.4 | 7.0 | 6.6 | 11.0 | 7.4 | 7.3 | 6.6 | 1.6 | 0.8 | 0.1 | 1.6 |
| LLaVA-TA (topic)* | 7B | **82.7** | **89.6** | **70.0** | **87.5** | **82.6** | **90.1** | **76.8** | **88.5** | **41.4** | **46.1** | **27.4** | **57.3** |
| LLaVA-TA (report)* | 7B | 79.6 | 89.2 | 61.9 | 87.2 | 79.5 | 89.1 | 69.6 | 87.7 | 31.1 | 41.1 | 22.5 | 38.6 |

### 4.1 TOPIC-GUIDED GENERATION IMPROVES CLINICAL ACCURACY

We first compare LLaVA-TA against state-of-the-art models on the MIMIC-CXR-JPG test set. As shown in Table. 1, our method establishes a new state of the art, significantly outperforming all baselines across both clinical accuracy (Smit et al., 2020; Irvin et al., 2019; Jain et al., 2021) and lexical similarity metrics (Papineni et al., 2002; Lin, 2004).

Most critically, LLaVA-TA demonstrates substantial gains in metrics that measure factual accuracy. Compared to the strong LLaVA-Rad baseline, our model improves the RadGraph F1 score from 29.4 to 34.3 and the CheXpert Macro-F1-14 from 39.5 to 62.4 at the report level. This significant improvement confirms that by decomposing the generation task into independent topics, our model produces more clinically precise and semantically correct findings. These results directly support

our central hypothesis: by preventing the model from relying on inter-sentence linguistic priors, we force it to ground its findings more strictly in visual evidence, leading to higher factual accuracy.

The superior performance on lexical metrics (e.g., BLEU-4 of 24.8 and ROUGE-L of 42.6) indicates that this enhanced accuracy does not come at the cost of linguistic quality. On the contrary, by focusing on one topic at a time, LLaVA-TA generates more precise and relevant language. Notably, our 7B parameter model also outperforms much larger, general-purpose models like Med-PaLM M (84B) and GPT-4V, highlighting that our structured, topic-guided supervision is a more efficient and effective strategy than simply scaling language model size.

The performance gap between the topic-level and report-level evaluations is expected. Metrics like RadGraph F1 score an entity as a false positive if the ground truth contains no mention of it, even if the model correctly generates a negative finding (e.g., "Heart is normal"). The topic-level evaluation avoids this by only assessing topics present in the ground truth, providing a cleaner measure of conditional generation quality. The results of LLaVA-Rad, LLaVA-T, and LLaVA-TA are the median of 500 bootstrap iterations. More detailed results can be found in Appendix. D and E.

## 4.2 TOPIC DISENTANGLEMENT UNLOCKS PERFORMANCE, WHILE ANATOMY ENHANCES EFFICIENCY

To isolate the contributions of our core design choices, we conduct a series of ablation studies. We compare our full LLaVA-TA model against LLaVA-T (which uses topic-guided decomposition but only the original image, without anatomical masks) and the LLaVA-Rad baseline.

**Topic Disentanglement is the Primary Driver of Performance.** As shown in Table 1, simply introducing topic-guided training (LLaVA-T) without anatomy masks already yields massive performance gains over the narrative-based LLaVA-Rad. RadGraph F1 improves from 29.4 to 34.2, and CheXpert Macro-F1-14 jumps from 39.5 to 63.0. This demonstrates that breaking the narrative flow is the single most critical factor in our model's success. It validates our hypothesis that forcing the model to generate independent, topic-specific findings is crucial for enhancing visual grounding and reducing hallucination.

**Anatomy-Awareness Enhances Parameter-Efficient Fine-Tuning** While adding anatomical masks (LLaVA-TA) provides comparable performance in the full fine-tuning setting, its true value emerges in a parameter-efficient fine-tuning (PEFT) regime. We conducted an experiment where only the multimodal MLP projector was trained, keeping the vision encoder and LLM frozen.

As shown in Table 2, under this MLP-only setting, LLaVA-TA dramatically outperforms both LLaVA-T and LLaVA-Rad. For instance, on MIMIC-CXR, LLaVA-TA achieves a RadGraph F1 of 31.1, whereas LLaVA-T and LLaVA-Rad lag far behind at 1.6 and 19.8, respectively. This suggests that when the LLM's parameters are frozen, the explicit spatial cues from the anatomical masks are vital. They provide a strong, structured signal that helps the lightweight MLP learn to map visual regions to the LLM's latent space effectively. This finding points to a promising direction for efficiently adapting large, frozen VLMs to the medical domain by using structured visual inputs.

Table 3: Out-of-distribution generalization performance on the IU-Xray dataset. LLaVA-TA maintains its significant performance advantage over the LLaVA-Rad baseline, especially on clinical accuracy metrics (RadGraph F1, CheXpert Macro-F1). This demonstrates the robustness of the topic-guided approach, which learns a stronger vision-language mapping less dependent on the linguistic priors of the training set.

| Model | Size | CheXpert | | | | | | | | RadGraph F1 | BLEU | | ROUGE-L |
| | | Micro-avg | | Macro-avg | | Micro-avg | | Macro-avg | | | (1) | (4) | |
| | | F1-14 | F1-5 | F1-14 | F1-5 | F1-14+ | F1-5+ | F1-14+ | F1-5+ | | | | |
| LLaVA-Rad | 7B | 50.3 | 38.9 | 28.8 | 29.9 | 48.2 | 39.3 | 30.9 | 33.5 | 19.8 | 32.0 | 6.4 | 22.2 |
| LLaVA-T (topic)* | 7B | **87.8** | **92.7** | **61.0** | **85.7** | **87.0** | **92.3** | 65.8 | **85.9** | 43.8 | **50.1** | 28.7 | **58.1** |
| LLaVA-T (report)* | 7B | 76.9 | 93.2 | 54.2 | 86.1 | 84.0 | 93.1 | 62.5 | 86.2 | 30.6 | 40.7 | 21.3 | 40.0 |
| LLaVA-TA (topic)* | 7B | 87.4 | 92.0 | 60.8 | 84.5 | 86.7 | 91.9 | **65.9** | 85.0 | 45.0 | 50.8 | 28.7 | 57.7 |
| LLaVA-TA (report)* | 7B | 77.4 | 92.5 | 54.5 | 84.7 | 84.5 | 92.7 | 63.4 | 85.4 | 31.4 | 41.8 | 21.5 | 40.9 |

## 4.3 GENERALIZATION, ROBUSTNESS, AND INTERPRETABILITY

Finally, we assess whether LLaVA-TA's improvements are robust and whether the model's internal mechanisms align with our theory of improved visual grounding.

Table 4: Out-of-distribution performance on IU-Xray with MLP-only fine-tuning. LLaVA-TA maintains its substantial performance lead in this challenging generalization setting. The strong results confirm that the benefits of combining topic disentanglement with anatomy-aware inputs are robust and transfer effectively to new data domains, even with minimal model updates.

| Model | Size | CheXpert | | | | | | | | RadGraph F1 | BLEU | | ROUGE-L |
| | | Micro-avg | | Macro-avg | | Micro-avg | | Macro-avg | | | (1) | (4) | |
| | | F1-14 | F1-5 | F1-14 | F1-5 | F1-14+ | F1-5+ | F1-14+ | F1-5+ | | | | |
| LLaVA-Rad | 7B | 20.2 | 18.7 | 12.9 | 16.4 | 20.2 | 21.0 | 13.4 | 18.6 | 23.5 | 11.9 | 3.0 | 12.1 |
| LLaVA-T (topic)* | 7B | 43.5 | 47.2 | 25.2 | 38.0 | 44.0 | 51.2 | 29.5 | 42.9 | 7.3 | 3.8 | 0.3 | 7.8 |
| LLaVA-T (report)* | 7B | 39.8 | 55.8 | 20.2 | 33.4 | 39.7 | 58.0 | 24.1 | 40.4 | 6.8 | 2.5 | 0.3 | 4.4 |
| LLaVA-TA (topic)* | 7B | **88.4** | **93.8** | **62.9** | **89.0** | **87.6** | **93.5** | **67.8** | **88.8** | **48.2** | **54.2** | **32.2** | **61.4** |
| LLaVA-TA (report)* | 7B | 77.9 | 94.3 | 57.6 | 88.5 | 85.4 | 94.2 | 66.0 | 88.4 | 34.0 | 43.5 | 24.7 | 43.4 |

**Strong Generalization to Out-of-Distribution Data.** We evaluated our model on the IU-Xray dataset, a common out-of-distribution benchmark. As detailed in Table 3 and 4, LLaVA-TA demonstrates superior generalization compared to the strong baseline and our topic-only LLaVA-T ablation study under both in-distribution and out-of-distribution settings. This large margin suggests that by reducing the reliance on dataset-specific linguistic priors (i.e., narrative flow), our topic-guided method learns a more fundamental and robust mapping between visual evidence and clinical findings, allowing it to adapt more effectively to new data distributions.

**Interpretability and Visual Grounding Analysis.** To validate the spatial fidelity of LLaVA-TA, we visualize its cross-attention maps during generation. As illustrated in Fig. 3, the model demonstrates consistent focus on relevant anatomical regions when generating topic-specific findings. For instance, when describing cardiac features, attention is strictly concentrated on the cardiac silhouette. This qualitative evidence confirms that our topic-guided strategy successfully fosters genuine spatial grounding, ensuring findings are derived from visual signals rather than textual priors.

We further analyze a challenging case in Fig. 4 where the generated report diverges textually from the ground truth. While the reference report notes "bibasal opacities," LLaVA-TA generates "a left retrocardiac opacity is present." Despite this linguistic mismatch, the attention visualization accurately highlights the posterior lower lung zones. This indicates that the model is not hallucinating; rather, it is maintaining internal consistency by grounding its specific prediction in valid visual features. This transparency is vital for clinical trust, as it allows radiologists to distinguish between model hallucinations and valid alternative interpretations of ambiguous visual data.

To move beyond qualitative inspection, we perform a rigorous quantification of attention alignment using the MS-CXR dataset (Boecking et al., 2022), which is a subset of the MIMIC-CXR-JPG dataset. MS-CXR is a phrase grounding benchmark containing curated image regions annotated with radiologist-verified descriptions of eight pathology categories. We evaluate on all 176 image-annotation pairs from the test subset using three metrics: (1) **Hit Score (Hit)** (Saporta et al., 2022), which credits the model if the maximum attention point falls within the ground-truth bounding box (BBox); (2) **Contrast-to-noise ratio (CNR)** (Boecking et al., 2022), defined as $CNR = |\mu_A - \mu_{\overline{A}}|/\sqrt{\sigma_A^2 + \sigma_{\overline{A}}^2}$, where $A$ and $\overline{A}$ represent regions inside and outside the BBox of the attention map, respectively; and (3) **Soft Precision (SP)** (Boecking et al., 2022), calculated as $SP = (\sum_{(x,y)\in BBox} S(x,y))/(\sum_{all(x,y)} S(x,y))$, where $S(x,y)$ is the attention map value at the position $(x,y)$. We also report $SP$-$k$, where attention maps are binarized by retaining the top $k\%$ of values before calculating precision.

The results, reported in Table 5 as the mean across all tokens within the sentence associated with the ground truth BBox, reveal a dramatic improvement. LLaVA-TA doubles the Hit Score of the baseline (from 29.2% to 59.2%) and achieves consistently higher CNR and Precision scores. This quantitative leap confirms that decomposing reports by topic significantly reduces the model's reliance on spurious correlations, forcing it to attend to the correct anatomical regions. Examples of visualization are provided in the Appendix F.

### 4.4 ABLATION STUDY

To quantify how much of the performance gain comes from dataset simplification and topic splitting, we compare the original dataset (OriData) against our topic-decomposed dataset (CleanData), and

Table 5: Attention map quantification on the MS-CXR benchmark. LLaVA-TA demonstrates superior visual grounding across all metrics, effectively doubling the Hit Rate compared to the baseline.

| Method | Hit(%)↑ | CNR(%)↑ | SP(%)↑ | SP-1(%)↑ | SP-3(%)↑ | SP-5(%)↑ | SP-10(%)↑ |
|---|---|---|---|---|---|---|---|
| LLaVA-Rad | 29.2 | 37.6 | 22.0 | 31.9 | 30.7 | 29.3 | 27.0 |
| **LLaVA-TA** | **59.2** | **43.4** | **37.5** | **62.0** | **55.7** | **50.3** | **41.4** |

compare standard narrative generation (allGTTopic, the concatenation of all the ground truth topics) against our proposed topic-guided generation (singleTopic, one topic at a time).

Table 6 shows that simply training on the cleaned data improves the RadGraph F1 score by only 1.6%. While this helps, it does not account for the majority of the performance leap. Transition from "CleanData" to "CleanData + singleTopic" yield a massive jump in RadGraph F1 from 32.4 to 46.5. This proves that the performance gain is primarily driven by our single-topic query mechanism, not by the simplified data format. The explicit querying forces the model to focus on factual accuracy rather than narrative style. Table 7 demonstrates the necessity of the anatomical mask when the LLM is frozen. Now, the model fails to generate accurate reports using the prompt alone ("CleanData + singleTopic", RadGraph F1 = 5.8). However, adding the mask ("+mask") restores performance to SOTA levels. This confirms that our "Anatomy-Aware" design is crucial for guiding the visual encoder when the language model capacity is constrained.

Table 6: Ablation study with MLP+LLM fine-tuning.

| Method | BLEU-1 | BLEU-4 | ROUGE-L | Micro-F1-14 | Macro-F1-14 | Micro-F1-5 | Macro-F1-5 | F1-RadGraph |
|---|---|---|---|---|---|---|---|---|
| OriData | 40.0 | 16.1 | 30.8 | 58.1 | 40.1 | 57.7 | 50.3 | 30.8 |
| CleanData | 46.4 | 26.2 | 40.0 | 63.3 | 40.7 | 66.0 | 57.3 | 32.4 |
| OriData + allGTTopic | 30.4 | 11.2 | 24.5 | 71.4 | 56.8 | 79.6 | 74.4 | 27.2 |
| CleanData + allGTTopic | 31.3 | 17.4 | 32.8 | 80.9 | 63.7 | 89.3 | 87.2 | 27.8 |
| **CleanData + singleTopic** | **50.9** | **32.1** | **60.9** | **83.7** | **72.2** | **90.0** | **87.9** | **46.5** |
| CleanData + singleTopic + mask | 50.9 | 31.9 | 60.7 | 83.6 | 71.6 | 89.6 | 87.3 | 46.3 |

Table 7: Ablation study with MLP only fine-tuning.

| Method | BLEU-1 | BLEU-4 | ROUGE-L | Micro-F1-14 | Macro-F1-14 | Micro-F1-5 | Macro-F1-5 | F1-RadGraph |
|---|---|---|---|---|---|---|---|---|
| OriData | 16.5 | 3.6 | 15.3 | 42.1 | 23.3 | 48.5 | 36.6 | 19.8 |
| CleanData | 9.9 | 1.4 | 10.8 | 31.8 | 17.7 | 35.1 | 24.2 | 10.4 |
| OriData + allGTTopic | 20.9 | 4.2 | 18.2 | 49.8 | 29.9 | 55.8 | 45.9 | 20.9 |
| CleanData + allGTTopic | 10.6 | 1.5 | 11.9 | 33.1 | 20.0 | 37.6 | 29.6 | 12.8 |
| CleanData + singleTopic | 2.9 | 0.4 | 5.9 | 27.7 | 20.4 | 27.5 | 22.3 | 5.8 |
| **CleanData + singleTopic + mask** | **46.1** | **27.4** | **57.3** | **82.7** | **70.0** | **89.6** | **87.5** | **41.4** |

## 5 DISCUSSION

In this work, we identified a fundamental limitation in existing approaches to Radiology Report Generation: an implicit optimization for narrative flow. We provided empirical evidence that this auto-regressive, sequential generation process encourages models to rely on linguistic priors at the expense of strict visual grounding, a primary cause of factual errors and clinical hallucination. To address this, we introduced LLaVA-TA, a novel framework that re-engineers report generation around a topic-guided paradigm. By decomposing the task into independent, anatomically-grounded clinical topics, LLaVA-TA breaks the cross-sentence dependency and forces the model to ground each finding directly in the relevant visual evidence. Our extensive experiments demonstrate that this approach sets a new SOTA on the MIMIC-CXR-JPG dataset in clinical accuracy (F1-RadGraph: $30.8 \rightarrow 46.5$, CheXpert Micro-F1-14: $58.1 \rightarrow 83.7$) and OOD generalization (F1-RadGraph on the unseen dataset IU-XRay: $19.8 \rightarrow 45.0$), proving that disentangling a report's narrative structure to enforce independent, visually-grounded findings is a crucial step toward building more reliable medical AI.

Despite the strong performance, our work has several limitations that open avenues for future research. First, our framework uses a static topic list; a dynamic approach that learns topics directly from data could capture more nuanced findings. Second, LLaVA-TA relies on an external segmentation model; an end-to-end architecture co-learning segmentation and generation could improve vision-language alignment. Finally, we currently exclude comparative sentences. Extending this framework to incorporate longitudinal temporal reasoning is critical for real-world clinical utility. LLaVA-TA provides a robust foundation for these future explorations.

## ETHICS STATEMENT

We acknowledge that all authors of this work have read and commit to adhering to the ICLR Code of Ethics.

## REPRODUCIBILITY STATEMENT

The processed dataset and model weights will be made publicly available on Hugging Face.

## ACKNOWLEDGEMENTS

Our research was funded by NIH 4R33-HD 105593. The content in the paper is solely the responsibility of the authors and does not necessarily represent the official views of the National Institutes of Health or the other funding agencies.

## THE USE OF LARGE LANGUAGE MODELS (LLMS)

During the preparation of this work the authors used ChatGPT in order to improve the readability and language of the manuscript. After using this tool, the authors reviewed and edited the content as needed and take full responsibility for the content of this article.

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

# A   DETAILED ANALYSIS OF THE NARRATIVE FLOW EXPERIMENT

Figure 5: Detailed performance scores for the narrative flow experiment. The plots show the absolute scores for both the oriImg and blackImg conditions across four metric categories. These results illustrate the model's behavior with and without visual input under varying degrees of textual context.

We first define "Narrative" Bias as follows: Narrative Bias is the tendency of an autoregressive model $P(y_t|y_{<t}, I)$ to maximize the likelihood of the next token $y_t$ based primarily on the linguistic probability $P(y_t|y_{<t})$ rather than the visual conditional probability $P(y_t|I)$.

To provide empirical evidence for our central hypothesis—that optimizing for narrative flow weakens a model's visual grounding—we conducted a controlled experiment. This section details the experimental setup and provides an in-depth analysis of the results.

## A.1   EXPERIMENTAL SETUP

We used a pretrained LLaVA-Rad model to generate the final K sentences of a radiology report, given the preceding sentences as a textual prompt. This task was performed under two distinct conditions:

- **oriImg** Condition: The model was provided with both the textual prompt and the corresponding ground-truth chest X-ray.
- **blackImg** Condition: The model was provided with the same textual prompt but with a blank (black) image, forcing it to rely solely on linguistic priors.

Denote the number of sentences for a ground truth report is $L$. We split the front $L - K$ sentences to the prompt and let the model generate the last $K$ sentences. We varied the amount of textual context by setting K to 3, 2, and 1 (denoted as $lastk - 3, lastk - 2, lastk - 1$). We also evaluated a baseline NoPrior condition where the model generated the full report from scratch, $K = L$. The performance under all settings is shown in Fig. 5.

## A.2   DETAILED PERFORMANCE ANALYSIS

The results in Figure 3 reveal several key trends that support our hypothesis: conventional training, by treating the report as a monolithic text sequence, inadvertently teaches models to prioritize linguistic coherence over strict visual fidelity. This learned reliance on the textual context of previously

generated sentences could become actively detrimental, allowing it to bias subsequent findings and override direct evidence from the image itself—a process that we suspect weakens visual grounding and fosters clinical hallucination.

We analyze the behavior within each condition below.

**Analysis of the oriImg condition**

- **Textual Context Degrades Visual Grounding:** The model achieves its highest scores in the NoPrior setting, where its generation is primarily guided by the visual evidence. The subsequent drop in performance across most metrics suggests that the preceding sentences introduce a strong linguistic bias, causing the model to generate text that follows the narrative, even at the expense of visual accuracy.

- **Strong Context Leads to Overfitting on Common Diseases:** A notable outlier occurs in the CheXpert-14 F1 score at $lastk - 1$. The Micro-F1 score (which is weighted by label frequency) remains high, while the Macro-F1 score (unweighted) drops sharply. This divergence indicates that when given strong textual context and a short generation target, the model defaults to predicting common, high-frequency diseases that are statistically likely to follow the prompt. It relies on learned textual correlations rather than observing the image, leading to overfitting on common pathologies.

**Analysis of the blackImg Condition: Textual Context as a Driver for Hallucination**

In the absence of visual input, the model's generation is driven entirely by the textual prompt. This setting allows us to directly measure how linguistic context induces hallucination.

- **Textual Priors Trigger Plausible, Structured Hallucinations:** The most direct evidence of hallucination is the jump in RadGraph F1 score when any textual context is provided (lastk settings) compared to none (blackImg NoPrior). RadGraph requires both a clinical finding and its anatomical location (e.g., "opacity" in the "lungs"). The model's ability to generate structurally correct entities from text alone demonstrates that the preceding sentences trigger a chain of plausible, yet entirely fabricated, clinical statements. This is not random guessing; it is coherent, context-driven hallucination.

- **More Context Narrows Hallucinations and Reduces "Lucky Guesses":** While it may seem counter-intuitive, performance scores generally decrease as more textual context is provided (from $lastk - 3$ to $lastk - 1$). This is not because the model is hallucinating less, but because the evaluation becomes stricter. A longer prompt creates a more specific narrative context, forcing the model to generate a more specific (and thus more likely to be incorrect) hallucination. When generating three sentences ($lastk - 3$), there is more opportunity for a random part of the hallucinated text to overlap with one of the three ground-truth sentences—a "lucky guess." When generating only one sentence ($lastk - 1$), the model must match a single, specific target, and any deviation results in a score of zero.

- **For Common Diseases, Textual Context Overwrites Visual Evidence:** The most striking finding is the trend of the Micro-F1-14 score in the blackImg condition. Unlike other metrics, this score increases as more context is provided, eventually matching the performance of the oriImg condition at the $lastk - 1$ setting. This provides definitive proof of our hypothesis. The Micro-F1 score is weighted by the prevalence of each disease label. The rising score shows that with strong textual priors, the model becomes highly proficient at hallucinating the most common diseases, to the point where the actual image provides no additional benefit. The linguistic patterns learned from the training data are so strong for these common cases that they completely override the need for visual input.

## B  TOPIC-SEGMENTATION MAPPING

This mapping is manually constructed between topics at the level of disease, organ, and other to the seven segmentation categories provided by the CXAS(Seibold et al., 2023) system.

Table 8: Mapping between topics and segmentation labels

| Topic | Segmentation labels |
|---|---|
| disease-atelectasis | lung halves |
| disease-cardiomegaly | heart region |
| disease-consolidation | lung halves |
| disease-edema | lung halves |
| disease-enlarged cardiomediastinum | mediastinum |
| disease-fracture | ribs, all vertebrae |
| disease-lung lesion | lung halves |
| disease-lung opacity | lung halves |
| disease-pleural effusion | lung halves |
| disease-pleural other | lung halves |
| disease-pneumonia | lung halves |
| disease-pneumothorax | lung halves |
| organ-heart | heart region |
| organ-lungs | lung halves |
| organ-pleura | lung halves |
| organ-mediastinum | mediastinum |
| organ-bones | ribs, all vertebrae |
| organ-diaphragm | diaphragm |
| support devices | all |
| patient status | all |
| other | all |

## C    PROMPT FOR SPLITTING REPORTS INTO SENTENCES BASED ON TOPIC

You are an AI radiology assistant specialized in preprocessing chest X-ray reports for vision-language model training. Your task is to extract and reorganize findings into standardized sentences, each aligned with a specific topic, while ensuring clarity and suitability for paired image-text learning.

**Rules for Processing Reports:**

1. **Hierarchical Topic Classification**
   - Classification Priority: Assign each sentence to the most specific applicable topic using this order:
   1.1 **Pathology-Level Topics:** disease-atelectasis, disease-cardiomegaly, disease-consolidation, disease-edema, disease-enlarged cardiomediastinum, disease-fracture, disease-lung lesion, disease-lung opacity, disease-pleural effusion, disease-pleural other, disease-pneumonia, disease-pneumothorax.
   1.2 **Anatomy-Level Topics** (Only if no pathology match): organ-heart, organ-lungs, organ-pleura, organ-mediastinum, organ-bones, organ-diaphragm.
   1.3 **Special Categories** (Only if no previous topics matches): support devices, patient status, other.
   - If a sentence covers multiple topics, split it into separate sentences (e.g., "Cardiomegaly and pulmonary edema" should be split into "Cardiomegaly is present." and "Pulmonary edema is present.")
   - **support devices:** Implanted or external hardware directly visible on imaging (e.g., pacemakers, catheters, surgical clips). Exclude historical procedures without visible remnants.
   - **patient status:** Clinically relevant history (e.g., post-surgical changes, intubation) that may explain imaging findings but does not describe visible devices/anatomy.
   - **other** for sentences that don't fit predefined topics but are visually relevant.

2. **Single Finding per Sentence:**
   - Each sentence must describe only one finding/disease (e.g., "The heart is enlarged with pleural effusion" should be split into "The heart is enlarged." and "Pleural effusion is present.")

3. **Handle Comparisons:**
   - If a sentence references prior images (e.g., "Increased opacity compared to prior"), reformat it to describe the current study only.
   - Example:
     - Before: "The pleural effusion has increased since last study."
     - After: "Pleural effusion is present."

4. **Exclude Non-Visual Content**
   - Remove the following:
     - Speculation/Interpretation: e.g., "suggestive of pneumonia", "likely due to CHF"
     - Recommendations: e.g., "Recommend follow-up CT."
     - Technical Artifacts: e.g., "Left lung is hard to see in the image."

5. **Negations and Absence**
   - Keep negative findings but phrase them affirmatively:
   - Example:
     - Before: "No pneumothorax is seen."
     - After: "Pneumothorax is absent."

6. **Clarity**
   - Use short, concrete phrases that directly map to visual features.
   - Examples:
     - Bad example: "There are some mild interstitial markings."

    – Good example: "Mild interstitial opacities are present."

**Output Format:**

[ "topic": "disease-lung opacity", "sentences": ["Substantial hyperexpansion of the lungs is present."], "topic": "organ-lungs", "sentences": ["Prominence of central pulmonary arteries is present.", "Severe emphysema is present."], "topic": "support devices", "sentences": ["A tracheostomy tube is present."] ]

Only output the JSON list without additional descriptions.

# D  PERFORMANCE OF FINE-TUNED MLP AND LLM

The following are the complete performance results for our model LLaVA-TA and the strong baseline model LLaVA-Rad under the settings of fine-tuning both the MLP and the language model.

Following LLaVA-Rad Zambrano Chaves et al. (2025), we obtain the median and 95% bootstrap confidence intervals from 500 resampling iterations.

Table 9: Detailed performance on the MIMIC test set when fine-tuning both MLP and LLM

| Model | Statistic | CheXpert | | | | | | | | RadGraph F1 | BLEU | | ROUGE-L |
| | | Micro-avg | | Macro-avg | | Micro-avg | | Macro-avg | | | (1) | (4) | |
| | | F1-14 | F1-5 | F1-14 | F1-5 | F1-14+ | F1-5+ | F1-14+ | F1-5+ | | | | |
| LLaVA-Rad | median | 58.1 | 57.7 | 40.1 | 50.3 | 57.7 | 59.8 | 42.9 | 53.0 | 30.8 | 40.0 | 16.1 | 30.8 |
| | CI-low | 57.2 | 56.3 | 38.6 | 48.6 | 56.9 | 58.6 | 41.5 | 51.6 | 30.2 | 39.4 | 15.6 | 30.3 |
| | CI-high | 59.0 | 59.1 | 41.5 | 52.1 | 58.6 | 61.0 | 44.2 | 54.4 | 31.4 | 40.5 | 16.5 | 31.3 |
| LLaVA-T (Topic)* | median | 83.7 | 90.0 | 72.2 | 87.9 | 83.7 | 90.3 | 78.2 | 88.7 | 46.5 | 50.9 | 32.1 | 60.9 |
| | CI-low | 83.2 | 89.5 | 70.7 | 87.2 | 83.2 | 89.8 | 77.1 | 88.1 | 45.9 | 50.3 | 31.6 | 60.5 |
| | CI-high | 84.2 | 90.6 | 73.6 | 88.6 | 84.3 | 90.9 | 79.3 | 89.4 | 47.2 | 51.5 | 32.7 | 61.4 |
| LLaVA-T (Report)* | median | 80.2 | 88.5 | 63.0 | 87.0 | 81.6 | 88.7 | 71.4 | 87.7 | 34.2 | 43.9 | 25.7 | 41.8 |
| | CI-low | 79.5 | 87.8 | 61.4 | 86.2 | 81.0 | 88.1 | 69.9 | 86.9 | 33.8 | 43.4 | 25.3 | 41.4 |
| | CI-high | 80.8 | 89.1 | 64.7 | 87.7 | 82.1 | 89.3 | 73.0 | 88.3 | 34.7 | 44.3 | 26.1 | 42.2 |
| LLaVA-TA (Topic)* | median | 83.5 | 89.6 | 71.5 | 87.3 | 83.6 | 90.1 | 77.9 | 88.3 | 44.0 | 50.9 | 31.8 | 60.6 |
| | CI-low | 83.0 | 89.0 | 70.1 | 86.5 | 83.1 | 89.5 | 76.8 | 87.5 | 43.4 | 50.3 | 31.3 | 60.2 |
| | CI-high | 84.1 | 90.2 | 73.0 | 88.0 | 84.2 | 90.6 | 78.9 | 89.0 | 44.6 | 51.5 | 32.4 | 61.2 |
| LLaVA-TA (Report)* | median | 80.0 | 89.0 | 62.4 | 86.8 | 81.0 | 88.7 | 70.7 | 87.4 | 34.3 | 43.7 | 24.8 | 42.6 |
| | CI-low | 79.3 | 88.4 | 60.8 | 86.0 | 80.5 | 88.1 | 69.3 | 86.7 | 33.8 | 43.3 | 24.4 | 42.3 |
| | CI-high | 80.7 | 89.6 | 64.1 | 87.5 | 81.6 | 89.3 | 72.2 | 88.2 | 34.8 | 44.1 | 25.2 | 43.0 |

Table 10: Detailed performance on the IU-Xray dataset when fine-tuning both MLP and LLM

| Model | Statistic | CheXpert | | | | | | | | RadGraph F1 | BLEU | | ROUGE-L |
| | | Micro-avg | | Macro-avg | | Micro-avg | | Macro-avg | | | (1) | (4) | |
| | | F1-14 | F1-5 | F1-14 | F1-5 | F1-14+ | F1-5+ | F1-14+ | F1-5+ | | | | |
| LLaVA-Rad | median | 50.3 | 38.9 | 28.8 | 29.9 | 48.2 | 39.3 | 30.9 | 33.5 | 19.8 | 32.0 | 6.4 | 22.2 |
| | CI-low | 49.6 | 37.3 | 27.6 | 27.9 | 47.5 | 37.9 | 29.8 | 31.7 | 19.6 | 31.8 | 6.3 | 22.1 |
| | CI-high | 51.0 | 40.4 | 30.0 | 32.2 | 48.9 | 40.7 | 32.1 | 35.2 | 20.0 | 32.2 | 6.5 | 22.3 |
| LLaVA-T (topic)* | median | 87.8 | 92.7 | 61.0 | 85.7 | 87.0 | 92.3 | 65.8 | 85.9 | 43.8 | 50.1 | 28.7 | 58.1 |
| | CI-low | 87.3 | 92.0 | 59.3 | 84.2 | 86.5 | 91.7 | 64.2 | 84.4 | 43.1 | 49.5 | 28.2 | 57.6 |
| | CI-high | 88.2 | 93.3 | 62.5 | 87.1 | 87.4 | 92.9 | 67.4 | 87.2 | 44.4 | 50.7 | 29.3 | 58.7 |
| LLaVA-T (report)* | median | 76.9 | 93.2 | 54.2 | 86.1 | 84.0 | 93.1 | 62.5 | 86.2 | 30.6 | 40.7 | 21.3 | 40.0 |
| | CI-low | 76.1 | 92.6 | 52.4 | 84.6 | 83.3 | 92.4 | 60.8 | 84.7 | 30.3 | 40.4 | 21.0 | 39.8 |
| | CI-high | 77.8 | 93.9 | 55.8 | 87.4 | 84.7 | 93.6 | 64.0 | 87.5 | 31.0 | 41.0 | 21.6 | 40.3 |
| LLaVA-TA (topic)* | median | 87.4 | 92.0 | 60.8 | 84.5 | 86.7 | 91.9 | 65.9 | 85.0 | 45.0 | 50.8 | 28.7 | 57.7 |
| | CI-low | 87.0 | 91.4 | 59.2 | 82.9 | 86.2 | 91.3 | 64.2 | 83.5 | 44.4 | 50.2 | 28.2 | 57.1 |
| | CI-high | 87.9 | 92.7 | 62.5 | 86.0 | 87.1 | 92.6 | 67.6 | 86.4 | 45.6 | 51.4 | 29.3 | 58.2 |
| LLaVA-TA (report)* | median | 77.4 | 92.5 | 54.5 | 84.7 | 84.5 | 92.7 | 63.4 | 85.4 | 31.4 | 41.8 | 21.5 | 40.9 |
| | CI-low | 76.6 | 91.9 | 52.9 | 83.2 | 83.8 | 92.1 | 61.7 | 84.0 | 31.1 | 41.5 | 21.3 | 40.6 |
| | CI-high | 78.2 | 93.2 | 55.9 | 86.1 | 85.1 | 93.3 | 65.0 | 86.9 | 31.8 | 42.1 | 21.8 | 41.2 |

# E  PERFORMANCE OF FINE-TUNED ONLY MLP

The following are the complete performance results for our model LLaVA-TA and the strong baseline model LLaVA-Rad under the settings of fine-tuning only the MLP.

Following LLaVA-Rad Zambrano Chaves et al. (2025), we obtain the median and 95% bootstrap confidence intervals from 500 resampling iterations.

Table 11: Detailed performance on the MIMIC test set when fine-tuning only MLP layers

| Model | Statistic | CheXpert | | | | | | | | RadGraph F1 | BLEU | | ROUGE-L |
| | | Micro-avg | | Macro-avg | | Micro-avg | | Macro-avg | | | (1) | (4) | |
| | | F1-14 | F1-5 | F1-14 | F1-5 | F1-14+ | F1-5+ | F1-14+ | F1-5+ | | | | |
|---|---|---|---|---|---|---|---|---|---|---|---|---|---|
| LLaVA-Rad | median | 42.1 | 48.5 | 23.3 | 36.6 | 40.6 | 48.8 | 25.1 | 39.0 | 19.8 | 16.5 | 3.6 | 15.3 |
| | CI-low | 41.0 | 47.0 | 22.3 | 35.1 | 39.4 | 47.3 | 24.1 | 37.5 | 19.4 | 16.3 | 3.5 | 15.1 |
| | CI-high | 43.4 | 50.1 | 24.4 | 38.3 | 41.7 | 50.2 | 26.2 | 40.6 | 20.2 | 16.8 | 3.8 | 15.5 |
| LLaVA-T (topic)* | median | 27.7 | 27.5 | 20.4 | 22.3 | 27.7 | 27.5 | 21.9 | 23.0 | 5.8 | 2.9 | 0.4 | 5.9 |
| | CI-low | 27.1 | 26.6 | 19.7 | 21.5 | 27.0 | 26.6 | 21.2 | 22.3 | 5.6 | 2.9 | 0.3 | 5.8 |
| | CI-high | 28.3 | 28.3 | 21.0 | 23.1 | 28.3 | 28.4 | 22.6 | 23.9 | 6.0 | 2.9 | 0.4 | 6.0 |
| LLaVA-T (report)* | median | 11.1 | 7.4 | 7.0 | 6.6 | 11.0 | 7.4 | 7.3 | 6.6 | 1.6 | 0.8 | 0.1 | 1.6 |
| | CI-low | 10.9 | 7.1 | 6.8 | 6.3 | 10.7 | 7.1 | 7.1 | 6.4 | 1.5 | 0.7 | 0.1 | 1.6 |
| | CI-high | 11.3 | 7.7 | 7.2 | 6.9 | 11.2 | 7.6 | 7.5 | 6.9 | 1.6 | 0.8 | 0.1 | 1.6 |
| LLaVA-TA (topic)* | median | 82.7 | 89.6 | 70.0 | 87.5 | 82.6 | 90.1 | 76.8 | 88.5 | 41.4 | 46.1 | 27.4 | 57.3 |
| | CI-low | 82.2 | 88.9 | 68.6 | 86.7 | 82.1 | 89.4 | 75.7 | 87.7 | 40.8 | 45.4 | 26.9 | 56.8 |
| | CI-high | 83.2 | 90.2 | 71.4 | 88.3 | 83.2 | 90.6 | 77.7 | 89.1 | 42.1 | 46.7 | 28.0 | 57.9 |
| LLaVA-TA (report)* | median | 79.6 | 89.2 | 61.9 | 87.2 | 79.5 | 89.1 | 69.6 | 87.7 | 31.1 | 41.1 | 22.5 | 38.6 |
| | CI-low | 78.9 | 88.6 | 60.4 | 86.5 | 79.0 | 88.6 | 68.2 | 87.1 | 30.6 | 40.5 | 22.1 | 38.3 |
| | CI-high | 80.2 | 89.8 | 63.4 | 87.9 | 80.1 | 89.7 | 71.0 | 88.4 | 31.6 | 41.8 | 23.0 | 39.0 |

Table 12: Detailed performance on the IU-Xray dataset when fine-tuning only MLP layers

| Model | Statistic | CheXpert | | | | | | | | RadGraph F1 | BLEU | | ROUGE-L |
| | | Micro-avg | | Macro-avg | | Micro-avg | | Macro-avg | | | (1) | (4) | |
| | | F1-14 | F1-5 | F1-14 | F1-5 | F1-14+ | F1-5+ | F1-14+ | F1-5+ | | | | |
|---|---|---|---|---|---|---|---|---|---|---|---|---|---|
| LLaVA-Rad | median | 20.2 | 18.7 | 12.9 | 16.4 | 20.2 | 21.0 | 13.4 | 18.6 | 23.5 | 11.9 | 3.0 | 12.1 |
| | CI-low | 19.5 | 17.6 | 12.2 | 15.0 | 19.6 | 20.0 | 12.9 | 17.5 | 23.4 | 11.8 | 3.0 | 12.1 |
| | CI-high | 20.8 | 19.8 | 13.5 | 17.6 | 20.7 | 22.1 | 14.0 | 19.7 | 23.7 | 11.9 | 3.0 | 12.2 |
| LLaVA-T (topic)* | median | 43.5 | 47.2 | 25.2 | 38.0 | 44.0 | 51.2 | 29.5 | 42.9 | 7.3 | 3.8 | 0.3 | 7.8 |
| | CI-low | 42.8 | 46.1 | 24.5 | 36.6 | 43.4 | 50.2 | 28.7 | 41.3 | 7.1 | 3.8 | 0.3 | 7.7 |
| | CI-high | 44.2 | 48.5 | 26.1 | 39.5 | 44.7 | 52.3 | 30.3 | 44.5 | 7.5 | 3.9 | 0.3 | 7.9 |
| LLaVA-T (report)* | median | 39.8 | 55.8 | 20.2 | 33.4 | 39.7 | 58.0 | 24.1 | 40.4 | 6.8 | 2.5 | 0.3 | 4.4 |
| | CI-low | 38.8 | 54.5 | 18.8 | 30.4 | 38.8 | 56.6 | 22.6 | 36.8 | 6.5 | 2.5 | 0.3 | 4.3 |
| | CI-high | 40.8 | 57.3 | 21.8 | 37.5 | 40.7 | 59.5 | 25.6 | 43.8 | 7.0 | 2.6 | 0.4 | 4.4 |
| LLaVA-TA (topic)* | median | 88.4 | 93.8 | 62.9 | 89.0 | 87.6 | 93.5 | 67.8 | 88.8 | 48.2 | 54.2 | 32.2 | 61.4 |
| | CI-low | 88.0 | 93.2 | 61.3 | 87.6 | 87.2 | 92.9 | 66.2 | 87.4 | 47.6 | 53.6 | 31.7 | 60.9 |
| | CI-high | 88.9 | 94.4 | 64.5 | 90.2 | 88.1 | 94.1 | 69.5 | 89.9 | 48.9 | 54.8 | 32.8 | 61.9 |
| LLaVA-TA (report)* | median | 77.9 | 94.3 | 57.6 | 88.5 | 85.4 | 94.2 | 66.0 | 88.4 | 34.0 | 43.5 | 24.7 | 43.4 |
| | CI-low | 77.0 | 93.7 | 55.7 | 87.0 | 84.7 | 93.7 | 64.2 | 87.0 | 33.6 | 43.2 | 24.4 | 43.1 |
| | CI-high | 78.7 | 94.8 | 59.5 | 89.8 | 86.2 | 94.8 | 67.6 | 89.6 | 34.4 | 43.8 | 25.0 | 43.7 |

## F  ATTENTION MAP QUANTIFICATION EXAMPLES

In this section, we present qualitative comparisons of attention maps generated by LLaVA-TA and LLaVA-Rad. To ensure a representative evaluation across the performance distribution, we stratify the test samples based on their Soft Precision scores (specifically using the top-3% binarization threshold, denoted as SP-3).

We categorize the results into three performance tiers for each model:

- **High Performance:** Samples falling within the top 30% of SP-3 scores.
- **Medium Performance:** Samples falling between the $30^{th}$ and $70^{th}$ percentiles.
- **Low Performance:** Samples falling within the bottom 30% of SP-3 scores.

We visualize one representative example from each tier in Fig. 6, Fig. 7, and Fig. 8, respectively. As observed across all three tiers, LLaVA-TA consistently produces more focused and anatomically precise attention maps compared to LLaVA-Rad. This qualitative evidence aligns with the quantitative results reported in Table 5, reinforcing the superior visual grounding capabilities of our proposed framework.

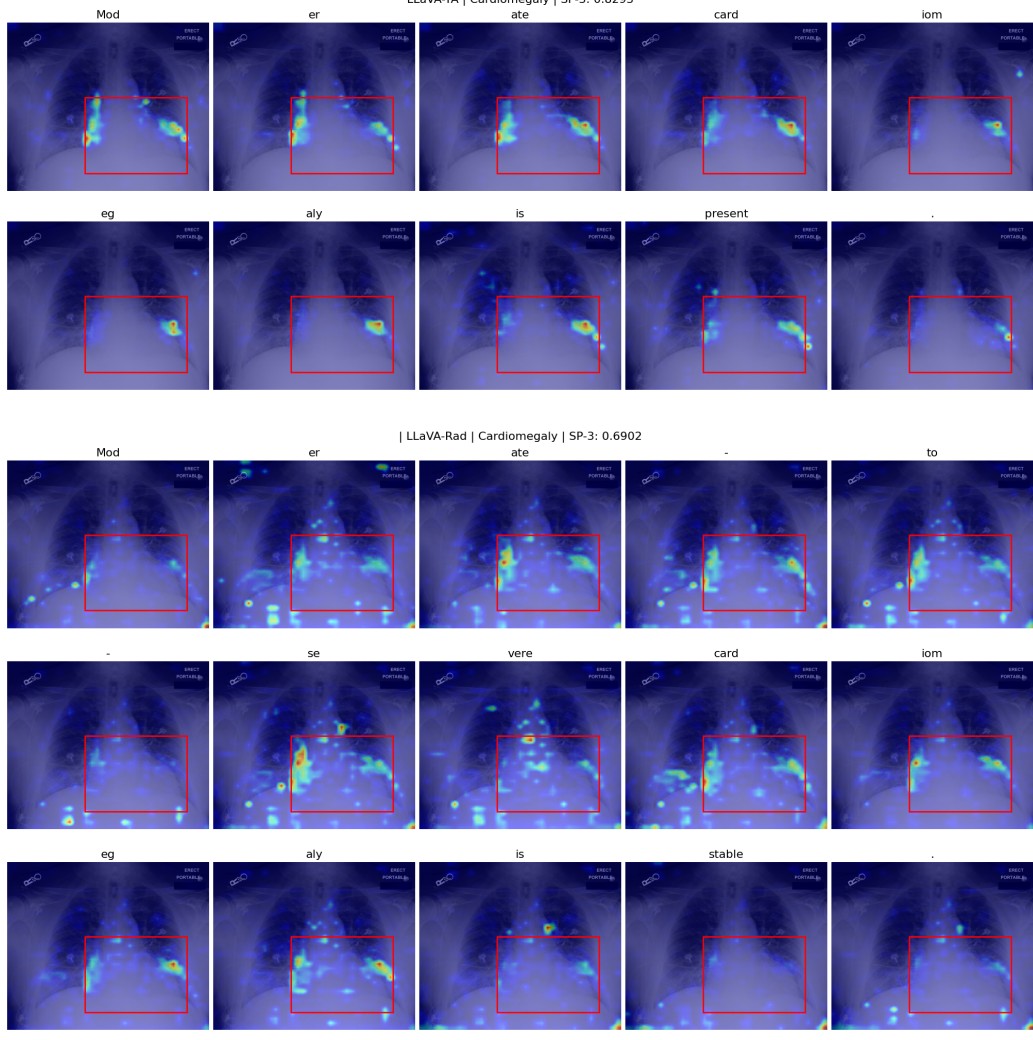

Figure 6: **High Performance Examples.** Attention map visualizations selected from the top 30% of SP-3 scores for each model.

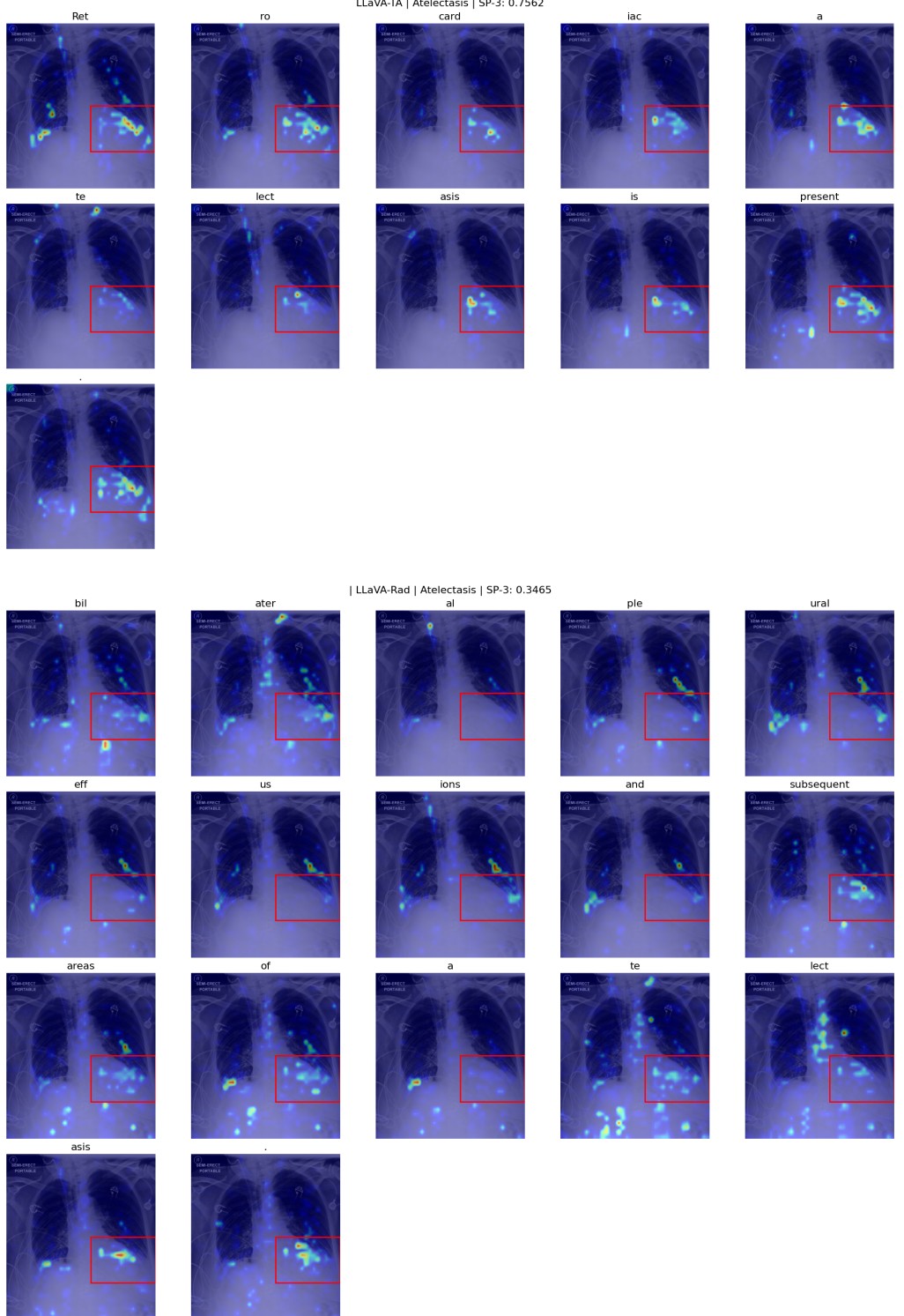

Figure 7: **Medium Performance Examples.** Attention map visualizations selected from the 30th to 70th percentile range of SP-3 scores.

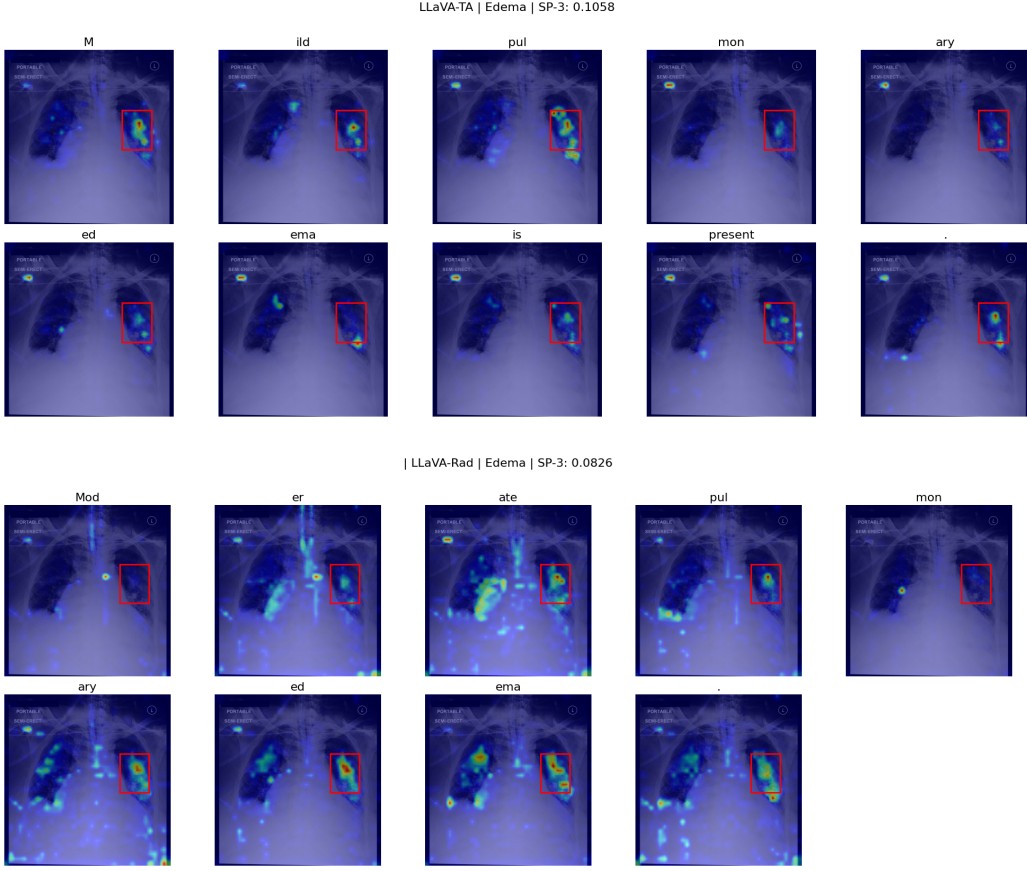

Figure 8: **Low Performance Examples.** Attention map visualizations selected from the bottom 30% of SP-3 scores.

# G    SENSITIVITY TO UPSTREAM ERRORS

## G.1    ROBUSTNESS TO SEMANTIC NOISE

We first analyzed the error modes of the DeepSeek-V3 pre-processing. We found the dominant error was omission (failing to extract a finding) rather than hallucination. To quantify our model's sensitivity to this, we conducted a Sensitivity Analysis via Topic Dropout.

We intentionally corrupted the training data by randomly dropping 1-2 topics from 5% of the training samples, simulating a scenario with significantly higher upstream omission rates.

Table 13: Evaluation of upstream errors (Topic split). We only train the MLP layers and keep both the vision encoder and the LLM frozen.

| Method | BLEU-1 | BLEU-4 | ROUGE-L | Micro-F1-14 | Macro-F1-14 | Micro-F1-5 | Macro-F1-5 | F1-RadGraph |
|---|---|---|---|---|---|---|---|---|
| LLaVA-Rad | 16.5 | 3.6 | 15.3 | 42.1 | 23.3 | 48.5 | 36.6 | 19.8 |
| LLaVA-TA+randomDrop | 45.4 | 27.0 | 57.1 | 82.7 | 69.7 | 89.5 | 87.3 | 40.9 |
| **LLaVA-TA** | **46.1** | **27.4** | **57.3** | **82.7** | **70.0** | **89.6** | **87.5** | **41.4** |

As shown in Table 13, the model trained on corrupted data shows negligible performance degradation (RadGraph F1: $41.4 \rightarrow 40.9$). This confirms that our framework is highly robust to upstream errors. Since our method treats each topic query independently, upstream omissions act merely as a reduction in dataset size rather than providing incorrect supervision.

## G.2    ROBUSTNESS TO VISUAL NOISE

To evaluate robustness against errors in the upstream segmentation module, we quantified the quality of all 224,105 segmentation masks based on their area ratios. We classified masks into **High** ($Z \leq 1$), **Mid** ($1 < Z < 2$), and **Low** ($Z \geq 2$) quality groups based on their deviation from the population mean, where $Z = \frac{\text{area\_ratio} - \mu}{\sigma}$, $\mu$ and $\sigma$ are the mean and standard deviation for the current segmentation mask's organ group.

(1) The analysis of the segmentation quality is done by calculating the relative mask area ratio across organs, as shown in Table 14.

Table 14: Analysis of segmentation quality by organ (area ratio statistics)

| Organ | Median | Q1 | Q3 | Mean | Std |
|---|---|---|---|---|---|
| Abdomen | 0.198 | 0.154 | 0.243 | 0.200 | 0.065 |
| All Vertebrae | 0.114 | 0.094 | 0.133 | 0.113 | 0.027 |
| Diaphragm | 0.241 | 0.192 | 0.292 | 0.243 | 0.073 |
| Heart Region | 0.083 | 0.072 | 0.096 | 0.084 | 0.019 |
| Lung Halves | 0.373 | 0.311 | 0.445 | 0.381 | 0.094 |
| Mediastinum | 0.159 | 0.139 | 0.180 | 0.160 | 0.030 |
| Ribs | 0.256 | 0.209 | 0.300 | 0.253 | 0.064 |

(2) Table 15 is the distribution of the three quality groups across organs.

Table 15: Distribution of quality groups across organs

| Organ | High ($Z \leq 1$) | Mid ($1 < Z < 2$) | Low ($Z \geq 2$) |
|---|---|---|---|
| Abdomen | 151,569 | 62,124 | 10,412 |
| All Vertebrae | 146,879 | 69,693 | 7,533 |
| Diaphragm | 151,038 | 62,980 | 10,087 |
| Heart Region | 156,214 | 54,983 | 12,908 |
| Lung Halves | 148,235 | 67,118 | 8,752 |
| Mediastinum | 150,300 | 64,145 | 9,660 |
| Ribs | 147,662 | 68,259 | 8,184 |

(3) Finally, the results of the three mask-quality groups are shown in Table 16 and 17.

Table 16 and 17 demonstrate two key findings:

**Correlation with Clinical Accuracy:** Crucially, we observe a positive correlation between mask quality and clinical classification performance (Micro-F1-14). In the MLP+LLM setting, the High-Quality group achieves 84.0, outperforming the Low-Quality group (80.1). This validates the effec-

Table 16: Model performance on different mask-quality groups with Fine-tuning MLP+LLM

| Group | BLEU-1 | BLEU-4 | ROUGE-L | Micro-F1-14 | Macro-F1-14 | Micro-F1-5 | Macro-F1-5 | F1-RadGraph |
|---|---|---|---|---|---|---|---|---|
| LLaVA-Rad | 40.0 | 16.1 | 30.8 | 58.1 | 40.1 | 57.7 | 50.3 | 30.8 |
| LLaVA-TA (low) | 52.1 | 32.5 | 60.3 | 80.1 | 56.8 | 91.5 | 86.8 | 43.0 |
| LLaVA-TA (mid) | 53.5 | 34.9 | 63.2 | 82.9 | 65.5 | 89.4 | 87.2 | 47.9 |
| LLaVA-TA (high) | 50.0 | 30.9 | 59.7 | 84.0 | 71.6 | 89.7 | 87.4 | 45.7 |
| LLaVA-TA (all) | 50.9 | 31.8 | 60.6 | 83.5 | 71.5 | 89.6 | 87.3 | 44.0 |

Table 17: Model performance on different mask-quality groups with Fine-tuning MLP only

| Group | BLEU-1 | BLEU-4 | ROUGE-L | Micro-F1-14 | Macro-F1-14 | Micro-F1-5 | Macro-F1-5 | F1-RadGraph |
|---|---|---|---|---|---|---|---|---|
| LLaVA-Rad | 16.5 | 3.6 | 15.3 | 42.1 | 23.3 | 48.5 | 36.6 | 19.8 |
| LLaVA-TA (low) | 47.9 | 26.6 | 55.6 | 77.5 | 53.3 | 90.8 | 88.6 | 37.0 |
| LLaVA-TA (mid) | 49.6 | 31.3 | 60.4 | 82.0 | 62.5 | 90.0 | 88.1 | 44.2 |
| LLaVA-TA (high) | 44.8 | 26.3 | 56.2 | 83.2 | 70.8 | 89.3 | 87.2 | 40.5 |
| **LLaVA-TA (all)** | **46.1** | **27.4** | **57.3** | **82.7** | **70.0** | **89.6** | **87.5** | **41.4** |

tiveness of our Anatomy-Aware design, demonstrating that accurate anatomical grounding directly contributes to diagnostic precision.

**Robustness via Soft Guidance:** Even on the "Low Quality" group (masks with extreme area deviations), the model achieves a RadGraph F1 of 37.0 (MLP setting), which is nearly 2x the performance of the LLaVA-Rad baseline (19.8). This confirms our Soft Guidance hypothesis: because the segmentation acts as a spatial prompt rather than a hard crop, the model retains access to the global context and remains effective even when masks are imperfect.

