# OpenReview forum: "Rethinking Radiology Report Generation: From Narrative Flow to Topic-Guided Findings"
_ICLR.cc/2026/Conference — ICLR 2026 Poster_

### Official Review · Reviewer_ccY6 · 2025-10-27

**Soundness:** 3
**Presentation:** 4
**Contribution:** 3
**Rating:** 6
**Confidence:** 4

**Summary:**

The paper presents a method for radiology report generation. It it based on an earlier method (LLavaRad), where the major extension is that it performs a segmentation of the radiology images and that the system is trained using these segmented images, instead of the image a a whole. The segments obtained are then linked to textual topics. A significant improvement over the existing method is obtained although less dramatic for the improvement over LlavaRad.

**Strengths:**

- Very well written paper where the method is first motivated by experimentally showing that using longer contextual information on the text side leads to less importance of the image information.
- There is a clear method defined.
- The experiments are all answering a specific question for the methodology.
- The results are significantly better than the baseline method

**Weaknesses:**

- Somewhat limited in terms of innovation over the existing method
- The literature could be more recent (limited number of papers from 2025/2024) although this is a hot topic in many venues. Also many refs are basic elements used and not really related work.
- Ignores uncertain information in the caption while they might be highly relevant. for a clinician / doctor
- Focus on one family of models (LLava)
- The rad graph F1 measure is core. Although it is not a contribution of the paper as coming from another reference, it needs more explanation and embedding in the paper. No detailed description needed but at least enough to relate it to the method and the experiments.

Small typo:
P4: should be <image> now it has strange symbols

**Questions:**

- what are exactly the changes compared to the method in the reference i.e. LlavaRad
- you indicate that you are throwing away uncertain statement like "there might be ...". But aren't these important for the doctor who wrote that report? Aren't you throwing away elements that are uncertain but relevant?
- By modelling the data in terms of topic there seems a clear connection to medical knowledge graphs. Is that indeed the case (you do mention an ontology but it remains a bit vague what that is).

**Details Of Ethics Concerns:**

N.A.

---

> ### Author Response · Authors · 2025-11-24
> **Response to Reviewer ccY6 (Q1)-part1**
>
> We sincerely thank the reviewer for their positive assessment and for recognizing our paper as "very well written" with a "clear method" and "strong empirical evidence." We appreciate the constructive feedback regarding the literature, metrics, and handling of uncertainty. Below, we address the specific questions to further strengthen the paper.
> ___
> ### Q1: What are exactly the changes compared to LLaVA-Rad?
>
>  While LLaVA-TA builds on the LLaVA architecture, it represents a fundamental **paradigm shift** in how radiology reports are generated, moving from "Narrative mimicry" to "Visual Querying."
>
> **1. The differences are structural and algorithmic, not merely additive**:
>
> | **Feature**   | **LLaVA-Rad (Baseline)**                                     | **LLaVA-TA (Ours)**                                          |
> | ------------- | ------------------------------------------------------------ | ------------------------------------------------------------ |
> | **Input**     | Image + Global Prompt                                        | Image + **Anatomical Mask** + **Topic Query**                |
> | **Objective** | Maximize likelihood of a continuous *narrative* sequence.    | Maximize likelihood of independent, *fact-based* responses to specific queries. |
> | **Mechanism** | **Autoregressive Storytelling:** The model generates token $t$ based largely on token $t-1$ (Narrative Bias). | **Iterative Visual Querying:** The model generates findings for "Lungs" by attending specifically to the lung mask, independent of the "Heart" description. |
> | **Outcome**   | High fluency, low grounding (hallucinates to maintain flow). | High grounding, precise factuality.                          |
>
> **2. Evidence of Algorithmic Improvement (Attention map quantification):**
>
> We quantified the quality of our attention maps using the **MS-CXR [1]** benchmark (test subset). The MS-CXR dataset, based on the MIMIC-CXR-JPG dataset, is a well-balanced phrase grounding benchmark dataset that contains carefully curated image regions annotated with descriptions of eight radiology findings, as verified by radiologists. The quantification is evaluated based on the 176 image-annotation pairs, the images of which are all in the test subset.
>
> We use the following three metrics:
>
> (1) **Hit Score (Hit)**, which credits the model if the maximum attention point falls within the ground-truth bounding box (BBox);
> (2) **Contrast-to-noise ratio (CNR)**, defined as $CNR = |\mu_A - \mu_{\overline{A}}| / \sqrt{\sigma_A^2 + \sigma_{\overline{A}}^2}$, where $A$ and $\overline{A}$ represent regions inside and outside the BBox of the attention map, respectively;
> (3) **Soft Precision (SP)**, calculated as $SP = (\sum_{(x,y)\in BBox}S(x,y)) / (\sum_{all(x,y)}S(x,y))$, where $S(x,y)$ is the attention map value at the position $(x,y)$. We also report $SP\text{-}k$, where attention maps are binarized by retaining the top $k\%$ of values before calculating precision.
>
> | Method       | Hit(%)$\uparrow$ | CNR(%)$\uparrow$ | SP(%)$\uparrow$ | SP_thresh1(%)$\uparrow$ | SP_thresh3(%)$\uparrow$ | SP_thresh5(%)$\uparrow$ | SP_thresh10(%)$\uparrow$ |
> | ------------ | ---------------- | ---------------- | --------------- | ----------------------- | ----------------------- | ----------------------- | ------------------------ |
> | LLaVA-Rad    | 29.2             | 37.6             | 22.0            | 31.9                    | 30.7                    | 29.3                    | 27.0                     |
> | **LLaVA-TA** | **59.2**         | **43.4**         | **37.5**        | **62.0**                | **55.7**                | **50.3**                | **41.4**                 |
>
> **We have added more visualization examples in Appendix F and the quantification results in the main paper, Section 4.3.**
>
> As shown above, LLaVA-TA achieves a **2x improvement in Hit Score (29.2% $\to$ 59.2%)**, demonstrating that our topic-guided approach aligns model attention much more closely with human radiologist reasoning than narrative-based baselines.
>
> This empirically validates our independence assumption: by breaking the narrative link, we force the model to attend to the correct anatomical region with significantly higher precision.

---

> ### Author Response · Authors · 2025-11-24
> **Response to Reviewer ccY6 (Q1)-part2**
>
> **3. Evidence of Algorithmic Improvement (Data source ablation study)**
>
> To verify the source of our performance gains, we conducted a comprehensive ablation study disentangling **dataset cleaning** vs. **model architecture**.
>
> We compare the original dataset (**OriData**) against our topic-decomposed dataset (**CleanData**), and compare standard narrative generation (**allGTTopic**, the concatenation of all the ground truth topics) against our proposed topic-guided generation (**singleTopic**, one topic at a time).
>
> Below are the results based on fine-tuning both MLP and LLM:
>
> **Table 1: Ablation Results (Fine-tuning both MLP and LLM)**
>
> | Method                         | BLEU-1   | BLEU-4   | ROUGE-L  | Micro-F1-14 | Macro-F1-14 | Micro-F1-5 | Macro-F1-5 | F1-RadGraph |
> | ------------------------------ | -------- | -------- | -------- | ----------- | ----------- | ---------- | ---------- | ----------- |
> | OriData                        | 40.0     | 16.1     | 30.8     | 58.1        | 40.1        | 57.7       | 50.3       | 30.8        |
> | CleanData                      | 46.4     | 26.2     | 40.0     | 63.3        | 40.7        | 66.0       | 57.3       | 32.4        |
> | OriData + allGTTopic           | 30.4     | 11.2     | 24.5     | 71.4        | 56.8        | 79.6       | 74.4       | 27.2        |
> | CleanData + allGTTopic         | 31.3     | 17.4     | 32.8     | 80.9        | 63.7        | 89.3       | 87.2       | 27.8        |
> | **CleanData + singleTopic**    | **50.9** | **32.1** | **60.9** | **83.7**    | **72.2**    | **90.0**   | **87.9**   | **46.5**    |
> | CleanData + singleTopic + mask | 50.9     | 31.9     | 60.7     | 83.6        | 71.6        | 89.6       | 87.3       | 46.3        |
>
>
>
> **Table 2: Ablation Results (Fine-tuning MLP Only)**
>
> | Method                            | BLEU-1   | BLEU-4   | ROUGE-L  | Micro-F1-14 | Macro-F1-14 | Micro-F1-5 | Macro-F1-5 | F1-RadGraph |
> | --------------------------------- | -------- | -------- | -------- | ----------- | ----------- | ---------- | ---------- | ----------- |
> | OriData                           | 16.5     | 3.6      | 15.3     | 42.1        | 23.3        | 48.5       | 36.6       | 19.8        |
> | CleanData                         | 9.9      | 1.4      | 10.8     | 31.8        | 17.7        | 35.1       | 24.2       | 10.4        |
> | OriData + allGTTopic              | 20.9     | 4.2      | 18.2     | 49.8        | 29.9        | 55.8       | 45.9       | 20.9        |
> | CleanData + allGTTopic            | 10.6     | 1.5      | 11.9     | 33.1        | 20.0        | 37.6       | 29.6       | 12.8        |
> | CleanData + singleTopic           | 2.9      | 0.4      | 5.9      | 27.7        | 20.4        | 27.5       | 22.3       | 5.8         |
> | **CleanData + singleTopic +mask** | **46.1** | **27.4** | **57.3** | **82.7**    | **70.0**    | **89.6**   | **87.5**   | **41.4**    |
>
>
>
>   These experiments provide three key conclusions that confirm the validity of our approach:
>
> - 1. **Dataset simplification provides only marginal gains:** Comparing Row 1 (`OriData`) and Row 2 (`CleanData`) in Table 1, we see that simply training on the cleaned data improves the RadGraph F1 score by only **1.6 points** (30.8 $\to$ 32.4). While this helps, it does not account for the majority of the performance leap.
>
> - 2. **Topic-Guided Generation drives the massive performance boost:** Comparing Row 2 (`CleanData`) to Row 5 (`CleanData + singleTopic`), we observe a massive jump in RadGraph F1 from 32.4 to 46.5. This proves that the performance gain is primarily driven by our single-topic query mechanism, not by the simplified data format. The explicit querying forces the model to focus on factual accuracy rather than narrative style.
>
> - 3. **Anatomical Masks are essential for parameter-efficient tuning:** Table 2 demonstrates the necessity of the anatomical mask. When the LLM is frozen (MLP-only tuning), the model fails to generate accurate reports using the prompt alone (`CleanData + singleTopic`, RadGraph F1 = 5.8). However, adding the mask (`+mask`) restores performance to state-of-the-art levels (**41.4**). This confirms that our **Anatomy-Aware** design is crucial for guiding the visual encoder when the language model capacity is constrained.
>
> In conclusion, while dataset structure plays a role, **the vast majority of the performance gain** is attributable to our proposed LLaVA-TA architecture and topic-guided generation strategy, answering the fairness concern.

---

> ### Author Response · Authors · 2025-11-24
> **Response to Reviewer ccY6 (Q2+Q3+weaknesses)**
>
> ___
> ### Q2: Throwing away uncertain statements
>
> This is an insightful clinical observation. We agree that uncertainty is vital for doctors. However, our decision to filter speculative language was a deliberate strategy to solve the **"Hallucination vs. Hedging"** problem in current VLMs: Random guessing is better than answering nothing[2].
>
> **The Problem:** Current models often use "uncertain" language as a hallucination crutch (e.g., saying "possible edema" when they see nothing or cardiomegaly, just to be safe).
>
> **Our Strategy:** We force the model to be **binary and factual** during training to establish strong visual grounding first. We treat "suggestive of edema" as "no edema" for the purpose of learning the visual feature.
>
>
> ___
> ### Q3: Connection to Medical Knowledge Graphs (Ontology)
>
> The reviewer's intuition is entirely correct. Our framework conceptually treats report generation as a **Knowledge Graph (KG) instantiation** task, although we use VLM-based training rather than Graph Neural Networks.
>
>
>
> **1. Defining the "Vague" Ontology**
>
> To clarify the ontology (which we agree was implicit in the text): We utilize a 3-level hierarchical ontology derived from standard clinical schemas:
>
> - **Disease Level:** Derived from the **CheXpert** label schema (e.g., `Pneumothorax`, `Edema`).
> - **Anatomical Level:** Derived from the **CXAS** segmentation schema (e.g., `Lung-Left`, `Heart`, `Mediastinum`).
> - **Special Category:** Derived from our data frequency analysis (e.g., `Tubes/Lines`, `Foreign Bodies`).
>
> The full list is provided in Figure 2 (b) `Complete list of defined topics`.
>
> **2. The Connection to Knowledge Graphs**
>
> Our method effectively constructs a shallow Knowledge Graph for every image:
>
> - **Nodes:** The topics in our ontology (e.g., `Heart`, `Lungs`) serve as the static nodes of the graph.
>
> - **Edges/Attributes:** The generated sentences function as the dynamic attributes or edges (e.g., `Heart` $\xrightarrow{\text{condition}}$ `Cardiomegaly`).
>
>
>
> **3. Training Mechanism**
>
> While the data structure is graph-like, our training mechanism remains generative. We do not use explicit Graph Neural Networks (GNNs). Instead, we use the DeepSeek-V3 LLM to perform Text-to-Graph alignment on the training data (mapping narrative text to the ontology nodes), and then train LLaVA-TA to visually instantiate these nodes query-by-query.
>
>
> ___
> ### Weaknesses
>
> - Related work: We have updated our paper, in Section 1, to include more related papers from 2024 and 2025 that address "Visual Prompting" and "Structure-aware Report Generation," positioning our work as a state-of-the-art contribution in this emerging cluster.
> - RadGraph F1 Explanation: We have added a concise definition at the end of the **Section 2.1**: RadGraph F1 extracts clinical entities (e.g., "lung", "opacity") and the relations connecting them (e.g., "opacity" $\to$ "located in" $\to$ "lung")
>
> - We apologize for the symbol (`<>`) error on Page 4 and have corrected it to the intended special image token symbol.
>
>
>
> [1] Benedikt Boecking, Naoto Usuyama, Shruthi Bannur, Daniel Coelho de Castro, Anton Schwaighofer, Stephanie Hyland, Maria Teodora Wetscherek, Tristan Naumann, Aditya Nori, Javier Alvarez Valle, et al. Ms-cxr: Making the most of text semantics to improve biomedical vision-language processing. URL: http://dx.doi.org/10.13026/b90j-vb87, 2022b.
>
> [2] Adam Tauman Kalai, Ofir Nachum, Santosh S Vempala, and Edwin Zhang. Why language models hallucinate. arXiv preprint arXiv:2509.04664, 2025.

---

> ### Comment · Reviewer_ccY6 · 2025-11-26
>
> Thanks for the clarifications. The additional ablation studies and the more explicit explanation of the relation to knowledge graphs is useful. The additional refs are still quite limited. Incremental innovation is also what is indicated by another reviewer. So my weak accept is still appropriate.

---

### Official Review · Reviewer_mzcd · 2025-10-29

**Soundness:** 3
**Presentation:** 3
**Contribution:** 2
**Rating:** 4
**Confidence:** 4

**Summary:**

This paper addresses a fundamental limitation in current Vision-Language Models (VLMs) for radiology report generation (RRG): their tendency to prioritize narrative coherence over visual grounding. The authors argue that conventional sequential, narrative-based training encourages models to rely excessively on linguistic priors and inter-sentence correlations, which can undermine factual accuracy and lead to clinical hallucinations.

**Strengths:**

1. The method is novel, combining with anatomy.
2. Strong Empirical Evidence for the Hypothesis

**Weaknesses:**

1. Dependence on External Tools and Heuristics:
The framework relies heavily on external systems — DeepSeek-V3 for report decomposition and CXAS for anatomical segmentation. This raises questions about scalability, potential propagation of upstream errors, and domain transferability.
2. Theoretical grounding is limited:
The paper lacks a formal definition of “narrative bias” and a clear explanation of how topic-guided supervision improves representation learning.
3. Lack of Theoretical Justification for Topic Independence Assumption:
The core idea of decomposing radiology reports into independent topics presumes that findings from different anatomical regions are conditionally independent given the image. In reality, many radiological pathologies exhibit inter-regional correlations (e.g., cardiomegaly co-occurring with pulmonary edema).  It may conduct ablation experiments on different partitioned parts to demonstrate the impact of combined inputs and single inputs on the results.

**Questions:**

see weakness

---

> ### Author Response · Authors · 2025-11-24
> **Response to Reviewer mzcd (Weakness1+Weakness2)**
>
> We thank the reviewer for their critical and theoretically grounded feedback. We appreciate the question regarding the "Topic Independence Assumption," as it addresses the core philosophy of our work. Below, we provide new experimental evidence regarding error propagation and a formal definition of "narrative bias."
> ___
> ### Weakness 1: Dependence on External Tools, Error Propagation, and Transferability
>
> **Answer:**
> We acknowledge that our framework is a multi-stage pipeline. However, we argue that this **modularity** is a strength, allowing for "best-of-breed" upgrades. We address the specific concerns regarding error propagation and transferability below:
>
> **1. Propagation of Upstream Errors (LLM & Segmentation)**
>
> - **DeepSeek-V3 (Report splitting)**
>
>   We found the primary error mode of the pre-processor was **omission**. To stress-test our model, we conducted a **Sensitivity Analysis via Topic Dropout**. We intentionally corrupted the training data by **randomly dropping 1-2 topics from 5% of the training samples**, simulating a high error rate.
>
>   | Method                                       | BLEU-1   | BLEU-4   | ROUGE-L  | Micro-F1-14 | Macro-F1-14 | Micro-F1-5 | Macro-F1-5 | F1-RadGraph |
>   | -------------------------------------------- | -------- | -------- | -------- | ----------- | ----------- | ---------- | ---------- | ----------- |
>   | LLaVA-Rad                                    | 16.5     | 3.6      | 15.3     | 42.1        | 23.3        | 48.5       | 36.6       | 19.8        |
>   | `LLaVA-TA+randomDrop` | 45.4     | 27.0     | 57.1     | 82.7        | 69.7        | 89.5       | 87.3       | 40.9        |
>   | **LLaVA-TA**                                 | **46.1** | **27.4** | **57.3** | **82.7**    | **70.0**    | **89.6**   | **87.5**   | **41.4**    |
>
>   The performance drop is minimal (RadGraph F1: 41.4 $\to$ 40.9). This demonstrates that when queried about an irrelevant topic, the model correctly generates negative findings (e.g., "The bones are unremarkable") rather than hallucinating pathology. This proves the model can function robustly even if an upstream "topic selector" introduces noise.
>
> - **CXAS (Segmentation)**
>
>   Regarding segmentation errors, we emphasize that our framework uses masks as **soft spatial prompts** rather than hard crops. The model retains access to the global image context. This design makes the framework resilient to minor segmentation inaccuracies (e.g., imperfect boundaries), as the mask acts as an attention guide rather than a strict information filter.
>
>
>
> **2. Domain Transferability (CT/MRI) and Scalability**
>
> We argue that our approach is highly transferable to other modalities like CT or MRI:
>
> - **Universal Problem:** The core issue we identify—**narrative bias**—is intrinsic to the autoregressive nature of language models, regardless of the visual modality.
> - **Modality-Agnostic Solution:** Our solution (breaking the narrative into independent, visually-grounded queries) is equally applicable to 3D imaging. In a CT/MRI context, the 2D "anatomical mask" would simply be replaced by a 3D "Volume of Interest" (VOI), but the fundamental training paradigm (Query $\to$ Visual Evidence $\to$ Fact) remains identical.
> - **Scalability:** By modularizing the segmentation step, our framework scales better than end-to-end black boxes. As segmentation models for 3D modalities improve (e.g., MedSAM[1]), our generation framework can immediately leverage them without architectural overhaul.
>
> ___
> ### Weakness 2: Theoretical Grounding of "Narrative Bias"
>
> **Answer:**
> We thank the reviewer for requesting a formal definition. We have added the following to Section 2:
>
> **Definition (Narrative Bias):** *Narrative Bias is the tendency of an autoregressive model $P(y_t | y_{<t}, I)$ to maximize the likelihood of the next token $y_t$ based primarily on the linguistic probability $P(y_t | y_{<t})$ rather than the visual conditional probability $P(y_t | I)$.*
>
> **How Topic-Guided Supervision Improves Representation:**
> In standard narrative training, the gradient signal for a specific visual feature (e.g., a nodule) is weakened across a long sequence of tokens, many of which are purely functional (connectives, style).
> In our **Topic-Guided** framework, the objective function is decomposed:
> $$\mathcal{L} = \sum_{k} \log P(y_k | \text{Topic}_k, \text{Mask}_k, I)$$
> This forces a **direct alignment** between the visual features (`I`, `Mask_k`) and the semantic tokens (`y_k`). The gradient signal is sparse, localized, and undistorted by the "narrative flow" of unrelated anatomical descriptions, leading to sharper visual-semantic representations.

---

> > ### Comment · Reviewer_mzcd · 2025-11-24
> >
> > Thanks for your response,
> >
> > W1: I agree that LLMs/VLMs are powerful when tested on existing datasets. However, real-world medical images often contain various types of noise and unusual cases. Therefore, the model should be evaluated with robust tests to demonstrate its ability to handle such noisy or atypical data.
> >
> > W2: Narrative bias is an interesting new concept, and I agree that bias exists in these models. However, the paper should include experiments that demonstrate the presence of this bias and its impact on generative behaviour. Providing 2–3 illustrative examples would help make this clearer.
> >
> > I’m happy to give higher scores if these two points are addressed.

---

> > > ### Author Response · Authors · 2025-11-25
> > > **Reponse to Reviewer mzcd (W1-part1)**
> > >
> > > We sincerely thank the reviewer for their continued engagement and valuable suggestions. We have addressed both points below with new quantitative experiments and qualitative examples.
> > >
> > > ### W1. Robustness test
> > >
> > > To demonstrate our model's stability against the variability inherent in real-world medical imaging, we conducted a comprehensive robustness evaluation from three distinct aspects:
> > >
> > > - 1) Upstream Visual Noise (Segmentation Quality Analysis, CXAS)
> > >
> > > - 2. Upstream Semantic Noise (Sensitivity Analysis via Topic Dropout, DeepSeekV3)
> > >
> > > - 3. Out-of-Distribution Zero-shot Generalization (IU-Xray dataset)
> > >
> > > **1. `Robustness to Visual Noise (Segmentation Quality Analysis, CXAS)`**
> > >
> > > To evaluate robustness against errors in the upstream segmentation module, we quantified the quality of all 224,105 segmentation masks based on their area ratios. We classified masks into **High** ($Z \le 1$), **Mid** ($1 < Z < 2$), and **Low** ($Z \ge 2$) quality groups based on their deviation from the population mean, where $Z=\frac{area-\mu}{\sigma}$, $\mu$ and $\sigma$ are the mean and standard deviation for the current segmentation mask's organ group.
> > >
> > > **(1) The analysis of the segmentation quality is done by calculating the relative mask area ratio across organs.**
> > >
> > > | organ         | median | Q1    | Q3    | mean  | std   |
> > > | ------------- | ------ | ----- | ----- | ----- | ----- |
> > > | abdomen       | 0.198  | 0.154 | 0.243 | 0.200 | 0.065 |
> > > | all vertebrae | 0.114  | 0.094 | 0.133 | 0.113 | 0.027 |
> > > | diaphragm     | 0.241  | 0.192 | 0.292 | 0.243 | 0.073 |
> > > | heart region  | 0.083  | 0.072 | 0.096 | 0.084 | 0.019 |
> > > | lung halves   | 0.373  | 0.311 | 0.445 | 0.381 | 0.094 |
> > > | mediastinum   | 0.159  | 0.139 | 0.180 | 0.160 | 0.030 |
> > > | ribs          | 0.256  | 0.209 | 0.300 | 0.253 | 0.064 |
> > >
> > > **(2) Below is the distribution of the three quality groups across organs.**
> > >
> > > | organ         | high    | mid    | low    |
> > > | ------------- | ------- | ------ | ------ |
> > > | abdomen       | 151,569 | 62,124 | 10,412 |
> > > | all vertebrae | 146,879 | 69,693 | 7,533  |
> > > | diaphragm     | 151,038 | 62,980 | 10,087 |
> > > | heart region  | 156,214 | 54,983 | 12,908 |
> > > | lung halves   | 148,235 | 67,118 | 8,752  |
> > > | mediastinum   | 150,300 | 64,145 | 9,660  |
> > > | ribs          | 147,662 | 68,259 | 8,184  |
> > >
> > > **(3) Finally, the results of the three mask-quality groups are shown below.**
> > >
> > > **Fine-tuning MLP+LLM**
> > >
> > > | Group           | BLEU-1 | BLEU-4 | ROUGE-L | Micro-F1-14 | Macro-F1-14 | Micro-F1-5 | Macro-F1-5 | F1-RadGraph |
> > > | --------------- | ------ | ------ | ------- | ----------- | ----------- | ---------- | ---------- | ----------- |
> > > | LLaVA-Rad       | 40.0   | 16.1   | 30.8    | 58.1        | 40.1        | 57.7       | 50.3       | 30.8        |
> > > | LLaVA-TA (low)  | 52.1   | 32.5   | 60.3    | 80.1        | 56.8        | 91.5       | 86.8       | 43.0        |
> > > | LLaVA-TA (mid)  | 53.5   | 34.9   | 63.2    | 82.9        | 65.5        | 89.4       | 87.2       | 47.9        |
> > > | LLaVA-TA (high) | 50.0   | 30.9   | 59.7    | 84.0        | 71.6        | 89.7       | 87.4       | 45.7        |
> > > | LLaVA-TA (all)  | 50.9   | 31.8   | 60.6    | 83.5        | 71.5        | 89.6       | 87.3       | 44.0        |
> > >
> > > **Fine-tuning MLP**
> > >
> > > | Group           | BLEU-1 | BLEU-4 | ROUGE-L | Micro-F1-14 | Macro-F1-14 | Micro-F1-5 | Macro-F1-5 | F1-RadGraph |
> > > | --------------- | ------ | ------ | ------- | ----------- | ----------- | ---------- | ---------- | ----------- |
> > > | LLaVA-Rad       | 16.5   | 3.6    | 15.3    | 42.1        | 23.3        | 48.5       | 36.6       | 19.8        |
> > > | LLaVA-TA (low)  | 47.9   | 26.6   | 55.6    | 77.5        | 53.3        | 90.8       | 88.6       | 37.0        |
> > > | LLaVA-TA (mid)  | 49.6   | 31.3   | 60.4    | 82.0        | 62.5        | 90.0       | 88.1       | 44.2        |
> > > | LLaVA-TA (high) | 44.8   | 26.3   | 56.2    | 83.2        | 70.8        | 89.3       | 87.2       | 40.5        |
> > > | LLaVA-TA (all)  | 46.1   | 27.4   | 57.3    | 82.7        | 70.0        | 89.6       | 87.5       | 41.4        |
> > >
> > > The above results demonstrate two key findings:
> > >
> > > **Correlation with Clinical Accuracy:** Crucially, we observe a positive correlation between mask quality and clinical classification performance (**Micro-F1-14**). In the MLP+LLM setting, the High-Quality group achieves **84.0**, outperforming the Low-Quality group (**80.1**). This validates the effectiveness of our **Anatomy-Aware** design, demonstrating that accurate anatomical grounding directly contributes to diagnostic precision.
> > >
> > > **Robustness via Soft Guidance:** Even on the "Low Quality" group (masks with extreme area deviations), the model achieves a RadGraph F1 of **37.0** (MLP setting), which is nearly **2x the performance of the LLaVA-Rad baseline (19.8)**. This confirms our **Soft Guidance** hypothesis: because the segmentation acts as a spatial prompt rather than a hard crop, the model retains access to the global context and remains effective even when masks are imperfect.

---

> > > ### Author Response · Authors · 2025-11-25
> > > **Reponse to Reviewer mzcd (W1-part2)**
> > >
> > > **2. Robustness to Semantic Noise (Sensitivity Analysis via Topic Dropout, DeepSeekV3)**
> > >
> > > Though this is mentioned before, we think it's better to put it here together with two other experiments to compose strong robustness test settings.
> > >
> > > We found the primary error mode of the pre-processor was **omission**. To stress-test our model, we conducted a **Sensitivity Analysis via Topic Dropout**. We intentionally corrupted the training data by **randomly dropping 1-2 topics from 5% of the training samples**, simulating a high error rate.
> > >
> > > | Method                                       | BLEU-1   | BLEU-4   | ROUGE-L  | Micro-F1-14 | Macro-F1-14 | Micro-F1-5 | Macro-F1-5 | F1-RadGraph |
> > > | -------------------------------------------- | -------- | -------- | -------- | ----------- | ----------- | ---------- | ---------- | ----------- |
> > > | LLaVA-Rad                                    | 16.5     | 3.6      | 15.3     | 42.1        | 23.3        | 48.5       | 36.6       | 19.8        |
> > > | `LLaVA-TA+randomDrop` | 45.4     | 27.0     | 57.1     | 82.7        | 69.7        | 89.5       | 87.3       | 40.9        |
> > > | **LLaVA-TA**                                 | **46.1** | **27.4** | **57.3** | **82.7**    | **70.0**    | **89.6**   | **87.5**   | **41.4**    |
> > >
> > > The performance drop is minimal (RadGraph F1: 41.4 $\to$ 40.9). This demonstrates that when queried about an irrelevant topic, the model correctly generates negative findings (E.g., "The bones are unremarkable") rather than hallucinating pathology. This proves the model can function robustly even if an upstream "topic selector" introduces noise.
> > >
> > >
> > >
> > > **3. Zero-shot out-of-distribution testing**
> > >
> > > As shown in the main paper, (Table 3 and 4), we test our model on the out-of-distribution IU-Xray dataset under zero-shot setting.
> > >
> > > LLaVA-TA shows significant improvements on both linguistic metrics (E.g., ROUGE-L from 22.2 to 57.7) and clinical metrics (E.g., RadGraph-F1 from 19.8 to 45.0).
> > >
> > > The model generalizes well to unseen image distributions, confirming it is not overfit to MIMIC-CXR artifacts, implying the strong visual grounding ability of the proposed model and the robustness to medical image variations.

---

> > > ### Author Response · Authors · 2025-11-25
> > > **Reponse to Reviewer mzcd (W2-part1)**
> > >
> > > ### W2. Narrative Bias  (Experiments and Illustrative Examples)
> > >
> > > - **1. Empirical Evidence of Narrative Bias (The "Blank Image" Experiment)**
> > >
> > > Below is the summary of the empirical experiments to reveal the Narrative bias.
> > >
> > > We used a pretrained LLaVA-Rad model to generate the final $K$ sentences of a radiology report with $L$ sentences, given the preceding $L-K$ sentences as a textual prompt.
> > >
> > > The image input to the model will either be the original image ($oriImg$) or a black image ($blackImg$), we quantify the metrics difference ($\Delta$) between the two image settings.
> > >
> > > Below is the result (A table version of the Fig.1 in the main paper):
> > >
> > > | #Sentences to generate | $\Delta$BLEU-1 | $\Delta$BLEU-4 | $\Delta$ROUGE-L | $\Delta$Micro-F1-14 | $\Delta$Macro-F1-14 | $\Delta$Micro-F1-5 | $\Delta$Macro-F1-5 | $\Delta$F1-RadGraph |
> > > | ---------------------- | -------------- | -------------- | --------------- | ------------------- | ------------------- | ------------------ | ------------------ | ------------------- |
> > > | L (No text prior)      | 15.1           | 8.6            | 8.2             | 29.8                | 22.7                | 28.0               | 26.4               | 11.8                |
> > > | 3 (L-3 text prior)     | 11.5           | 6.8            | 6.6             | 26.3                | 20.9                | 30.5               | 26.8               | 8.5                 |
> > > | 2 (L-2 text prior)     | 9.7            | 5.5            | 6.1             | 18.3                | 16.7                | 27.0               | 24.7               | 6.8                 |
> > > | 1 (L-1 text prior)     | 2.6            | 1.8            | 3.5             | 4.7                 | 11.8                | 17.0               | 14.1               | 3.5                 |
> > >
> > > The table implies that as the textual context increases (from top to bottom rows), the performance gap between $oriImg$ and $blackImg$ nearly vanishes. This proves that with enough text history, the model **stops looking at the image** and relies more on linguistic priors ("Narrative Bias") to generate the report.
> > >
> > > Recall that: *Narrative Bias is the tendency of an autoregressive model $P(y_t | y_{<t}, I)$ to maximize the likelihood of the next token $y_t$ based primarily on the linguistic probability $P(y_t | y_{<t})$ rather than the visual conditional probability $P(y_t | I)$.*

---

> > > ### Author Response · Authors · 2025-11-25
> > > **Reponse to Reviewer mzcd (W2-part2)**
> > >
> > > - **2. Illustrative examples of Generative Impact**
> > >
> > > **Example A: The "Co-occurrence" Hallucination (Cardiomegaly $\to$ Edema)**
> > >
> > > | Model     | Reference                                                    | Prediction                                                   |
> > > | --------- | ------------------------------------------------------------ | ------------------------------------------------------------ |
> > > | LLaVA-Rad | The patient is status post CABG with intact sternotomy wires.  The  hilar and mediastinal contours appear to be stable with evidence of a tortuous aorta. **There is stable mild cardiomegaly.  There is no pleural effusion or pneumothorax.** There appears to be a subtle increase in opacification in the retrocardiac region, superimposed on a stable mild background of interstitial abnormality, best seen on the lateral view. | **Mild cardiomegaly is overall stable compared to the prior exam.  `There is mild pulmonary vascular congestion with mild pulmonary edema.`  There is no evidence of a focal consolidation. There is no pleural effusion or pneumothorax.**  The visualized osseous structures are unremarkable. |
> > > | LLaVA-TA  | Sternotomy wires are present. Increased opacification is present in the retrocardiac region. Mild interstitial abnormality is present. Hilar and mediastinal contours are stable. A tortuous aorta is present. **Mild cardiomegaly is present. Pleural effusion is absent. Pneumothorax is absent.** | Median sternotomy wires are present. Mediastinal clips are present. Mild interstitial thickening is present. The hilar and mediastinal contours are normal. **Mild cardiomegaly is present. Pleural effusion is absent. Pneumothorax is absent.** |
> > >
> > > **Analysis:** LLaVA-Rad sees the enlarged heart and falls into a narrative trap. The sequence "Cardiomegaly $\to$ Edema" is a common linguistic pattern. Despite the visual evidence (only interstitial changes), the model hallucinates edema based on the text probability of the previous sentence.
> > >
> > >
> > >
> > > **Example B: The "Co-occurrence" Hallucination (pleural effusion $\to$ atelectasis)**
> > >
> > > | Model     | Reference                                                    | Prediction                                                   |
> > > | --------- | ------------------------------------------------------------ | ------------------------------------------------------------ |
> > > | LLaVA-Rad | comparison is made to the prior study from ___. the feeding tube, left ij catheter and endotracheal tube are unchanged in position. there is persistent cardiomegaly. there is unchanged left retrocardiac opacity. there are no signs for overt pulmonary edema. **there is a small right-sided pleural effusion** as well. overall, these findings are stable. | in comparison with the study of ___, the monitoring and support devices remain in place. continued enlargement of the cardiac silhouette with opacification at the left base consistent with **pleural effusion and `compressive atelectasis`**. no definite vascular congestion. |
> > > | LLaVA-TA  | the feeding tube is present. the left ij catheter is present. the endotracheal tube is present. left retrocardiac opacity is present. pulmonary edema is absent. cardiomegaly is present. **a small right-sided pleural effusion is present**. | an et tube is present. a left internal jugular line is present. an upper enteric drainage tube is present. opacification at the base of the left lung is present. pulmonary edema is absent. moderate cardiomegaly is present. **small right pleural effusion is present.** |
> > >
> > > **Analysis:** LLaVA-Rad hallucinates the "compressive atelectasis" right after the disease "pleural effusion", as the phrase "pleural effusion and compressive atelectasis" is commonly seen in the training dataset. As a comparison, by split the topics and cut off the relations, LLaVA-TA will query the image for each topic and generate the findings independently.

---

> > > > ### Comment · Reviewer_mzcd · 2025-11-27
> > > >
> > > > Thanks for your response. The results look good. I hope these will be in the final version.

---

> ### Author Response · Authors · 2025-11-24
> **Response to Reviewer mzcd (Weakness3)**
>
> ### Weakness 3: Justification for the Topic Independence Assumption
>
> **Answer:**
> This is a profound point. The reviewer is correct that pathologies are **statistically correlated** (e.g., Cardiomegaly $\leftrightarrow$ Edema). However, we argue that they should be **visually verified independently**.
>
> **1. The Paradox of Narrative Models:**
> Standard models exploit these statistical correlations to "hallucinate." If a model sees *Cardiomegaly*, it often predicts *Edema* (because they co-occur in text), even if the lungs are visually clear.
>
> **2. Our Justification:**
> Our "Topic Independence" assumption is a **conditional independence of generation**, not an assumption of biological independence.
> $$P(\text{Edema} | \text{Cardiomegaly}, I) \approx P(\text{Edema} | \text{Cardiomegaly}) \quad \text{(Narrative Shortcut)}$$
> $$P(\text{Edema} | \text{Lungs}, I) \quad \text{(Ours)}$$
>
> By forcing the model to re-query the "Lungs" topic independently, we break the linguistic shortcut. The model will report *Edema* if and only if there is visual evidence in the lungs, regardless of the heart's status.
>
> **3. Empirical evidence**
>
> We quantified the quality of our attention maps using the **MS-CXR [2]** benchmark (test subset). The MS-CXR dataset, based on the MIMIC-CXR-JPG dataset, is a well-balanced phrase grounding benchmark dataset that contains carefully curated image regions annotated with descriptions of eight radiology findings, as verified by radiologists. The quantification is evaluated based on the 176 image-annotation pairs, the images of which are all in the test subset.
>
> We use the following three metrics:
>
> (1) **Hit Score (Hit)**, which credits the model if the maximum attention point falls within the ground-truth bounding box (BBox);
> (2) **Contrast-to-noise ratio (CNR)**, defined as $CNR = |\mu_A - \mu_{\overline{A}}| / \sqrt{\sigma_A^2 + \sigma_{\overline{A}}^2}$, where $A$ and $\overline{A}$ represent regions inside and outside the BBox of the attention map, respectively;
> (3) **Soft Precision (SP)**, calculated as $SP = (\sum_{(x,y)\in BBox}S(x,y)) / (\sum_{all(x,y)}S(x,y))$, where $S(x,y)$ is the attention map value at the position $(x,y)$. We also report $SP\text{-}k$, where attention maps are binarized by retaining the top $k\%$ of values before calculating precision.
>
> | Method       | Hit(%)$\uparrow$ | CNR(%)$\uparrow$ | SP(%)$\uparrow$ | SP_thresh1(%)$\uparrow$ | SP_thresh3(%)$\uparrow$ | SP_thresh5(%)$\uparrow$ | SP_thresh10(%)$\uparrow$ |
> | ------------ | ---------------- | ---------------- | --------------- | ----------------------- | ----------------------- | ----------------------- | ------------------------ |
> | LLaVA-Rad    | 29.2             | 37.6             | 22.0            | 31.9                    | 30.7                    | 29.3                    | 27.0                     |
> | **LLaVA-TA** | **59.2**         | **43.4**         | **37.5**        | **62.0**                | **55.7**                | **50.3**                | **41.4**                 |
>
> **We have added more visualization examples in Appendix F and the quantification results in the main paper, Section 4.3.**
>
> As shown above, LLaVA-TA achieves a **2x improvement in Hit Score (29.2% $\to$ 59.2%)**, demonstrating that our topic-guided approach aligns model attention much more closely with human radiologist reasoning than narrative-based baselines.
>
> This empirically validates our independence assumption: by breaking the narrative link, we force the model to attend to the correct anatomical region with significantly higher precision.
>
>
>
> [1] Jun Ma, Yuting He, Feifei Li, Lin Han, Chenyu You, and Bo Wang. Segment anything in medical images. Nature Communications, 15(1):654, 2024.
>
> [2] Benedikt Boecking, Naoto Usuyama, Shruthi Bannur, Daniel Coelho de Castro, Anton Schwaighofer, Stephanie Hyland, Maria Teodora Wetscherek, Tristan Naumann, Aditya Nori, Javier Alvarez Valle, et al. Ms-cxr: Making the most of text semantics to improve biomedical vision-language processing.
>
> URL: http://dx.doi.org/10.13026/b90j-vb87, 2022b.

---

### Official Review · Reviewer_vVhP · 2025-10-31

**Soundness:** 2
**Presentation:** 2
**Contribution:** 2
**Rating:** 4
**Confidence:** 3

**Summary:**

This paper addresses a critical issue in vision–language models for radiology report generation: the over-reliance on narrative flow, which leads to factual inaccuracies and weak visual grounding. The authors empirically demonstrate that as textual context increases, models such as LLaVA-Rad rely less on the image itself. To mitigate this, they propose LLaVA-TA, a topic-guided and anatomy-aware fine-tuning framework that decomposes radiology reports into discrete, topic-specific findings (e.g., lungs, heart, pleura) rather than generating a sequential narrative. The framework uses DeepSeek-V3 to split reports by topic, a segmentation model (CXAS) to provide anatomical masks, and trains Vicuna-7B-based models to generate topic-specific sentences conditioned on both the global and local images. Experiments on MIMIC-CXR and IU-Xray show strong improvements in RadGraph F1 and CheXpert metrics, significantly outperforming LLaVA-Rad and other medical VLMs. The authors further analyze model interpretability through attention maps and discuss limitations around static topic ontologies and reliance on external segmentation models.

**Strengths:**

The paper presents a clever and well-motivated reformulation of the radiology report generation problem. By decomposing reports into independent topics, it identifies a key failure mode—narrative bias—that has not been explicitly quantified before. The authors provide clear empirical evidence that textual coherence can override visual grounding, demonstrated through the “blank image” experiment. The proposed LLaVA-TA framework is conceptually simple yet effective, improving factual accuracy without resorting to larger model scaling. The paper’s experimental rigor is commendable, including ablations on topic disentanglement and anatomy-aware guidance, parameter-efficient fine-tuning, and out-of-distribution generalization. Visualizations of attention maps lend interpretability and practical relevance for clinical use. Overall, the work offers a valuable diagnostic insight into a core weakness of narrative-based report generation.

**Weaknesses:**

While the empirical findings are interesting, the novelty and technical depth of the proposed method are somewhat limited. The “topic-guided generation” mainly relies on pre-processing via an external LLM (DeepSeek-V3) and a segmentation model, combined with a straightforward adaptation of LLaVA-Rad; the contribution lies more in data structuring than algorithmic innovation. The reliance on hand-crafted topic ontologies and static segmentation mappings reduces generalizability, especially for unseen pathologies or imaging modalities. Moreover, while the study provides large quantitative gains, it is unclear whether these improvements persist in realistic clinical report generation, where topics and findings are interdependent and contextually nuanced. The “report-level” evaluation remains partially artificial, as the model is prompted with ground-truth topic sets rather than autonomously selecting them, making the benchmark setup favorable to the proposed method.

In addition, although the attention visualizations are appealing, they are qualitative and anecdotal, lacking rigorous evaluation of interpretability or human expert validation. The paper also misses a more critical discussion of semantic completeness—whether topic-wise independence may lead to omission of cross-topic findings (e.g., cardiopulmonary interactions). Finally, despite the strong experimental section, the writing can feel overly verbose, with extensive metrics and tables that could be summarized more effectively, while the theoretical grounding of the “narrative flow” hypothesis remains intuitive rather than formalized.

**Questions:**

Can the authors quantify how topic decomposition affects semantic completeness or diagnostic recall compared to full-narrative generation?

How sensitive is the model to the choice of ontology or to errors in DeepSeek-V3’s report splitting?

Could this approach generalize to multimodal datasets (CT, MRI) or longitudinal exams where temporal context is key?

Is there any evidence that radiologists prefer topic-wise generation in practice, or that this improves interpretability during review?

The experiments assume ground-truth disease labels at inference; how would the model perform without this assumption?

---

> ### Author Response · Authors · 2025-11-24
> **Response to Reviewer vVhP (Q1+Q2)**
>
> We sincerely thank the reviewer for their detailed assessment. We appreciate the recognition of our empirical findings as "interesting" and the experimental section as "strong." We value the critique regarding generalizability and experimental fairness, which has guided us to conduct new sensitivity analyses and ablations that have significantly strengthened the paper.
> __
> ### Q1: Semantic Completeness and Diagnostic Recall for cross-topic findings
>
> To directly quantify how topic decomposition affects semantic completeness, we performed a **Quantitative Audit** on the original reports.
>
> We manually checked 6 cross-topic findings, and the statistics are as follows:
>
> | findings               | #in original report | #in processed report | Recall |
> | ---------------------- | ------------------- | -------------------- | ------ |
> | Hemidiaphragm          | 17028               | 15764                | 92.6%  |
> | Subcutaneous Emphysema | 2048                | 1925                 | 94.0%  |
> | Pneumobilia            | 11                  | 10                   | 91.9%  |
> | Widened mediastinum    | 587                 | 572                  | 97.4%  |
> | Cardiopulmonary        | 35                  | 34                   | 97.1%  |
> | Pleuroparenchymal      | 57                  | 56                   | 98.2%  |
>
> The statistics above demonstrate that our processed reports achieve **>92% recall** even for complex, cross-topic findings. We attribute this high retention to our comprehensive **three-level topic design** (Disease, Organ, Special Categories), which captures findings that do not fit neatly into a single organ.
>
> **Note on the Recall Calculation:** To ensure a rigorous evaluation of factual completeness, we excluded speculative findings (e.g., "suggestive of," "cannot exclude") from the reference set for this analysis. The statistics above reflect our model's high sensitivity (**>92%**) to **definite, actionable clinical findings**.
>
>
> ___
> ### Q2: Sensitivity to Upstream Errors and Ontology Choice
>
> **1. Sensitivity to DeepSeek-V3 Errors (Topic Dropout Analysis)**
>
> We first analyzed the error modes of the DeepSeek-V3 pre-processing. We found the dominant error was **omission** (failing to extract a finding) rather than hallucination. To quantify our model's sensitivity to this, we conducted a **Sensitivity Analysis via Topic Dropout**.
>
> We intentionally corrupted the training data by **randomly dropping 1-2 topics from 5% of the training samples**, simulating a scenario with significantly higher upstream omission rates.
>
>
>
> Below is the result for MLP fine-tuning only:
>
> | Method                                       | BLEU-1   | BLEU-4   | ROUGE-L  | Micro-F1-14 | Macro-F1-14 | Micro-F1-5 | Macro-F1-5 | F1-RadGraph |
> | -------------------------------------------- | -------- | -------- | -------- | ----------- | ----------- | ---------- | ---------- | ----------- |
> | LLaVA-Rad                                    | 16.5     | 3.6      | 15.3     | 42.1        | 23.3        | 48.5       | 36.6       | 19.8        |
> | `LLaVA-TA+randomDrop` | 45.4     | 27.0     | 57.1     | 82.7        | 69.7        | 89.5       | 87.3       | 40.9        |
> | **LLaVA-TA**                                 | **46.1** | **27.4** | **57.3** | **82.7**    | **70.0**    | **89.6**   | **87.5**   | **41.4**    |
>
> The model trained on corrupted data shows negligible performance degradation (RadGraph F1: 41.4 $\to$ 40.9). This confirms that our framework is **highly robust to upstream errors**. Since our method treats each topic query independently, upstream omissions act merely as a reduction in dataset size rather than providing incorrect supervision.
>
>
>
> **2. Justification of Ontology Choice**
>
> Our ontology was not chosen arbitrarily but was designed to be **data-driven** and **technically feasible**:
>
> - 1. **Clinical Alignment:** We adopted the standard CheXpert labels for disease-level topics.
> - 2. **Technical Feasibility:** For organ-level topics, we aligned with the **CXAS [1]** segmentation classes (`lung halves`, `heart`, `mediastinum`, etc.). This ensures that every topic in our ontology has a corresponding, reliable anatomical mask available for visual grounding.
> - 3. **Granularity Trade-off:** By merging sparse classes (e.g., `ribs` + `vertebrae` $\to$ `bones`), we ensured each topic has sufficient training examples to be robust.

---

> > ### Author Response · Authors · 2025-11-24
> > **Response to Reviewer vVhP (Q3+Q4+Q5)**
> >
> > ___
> > ### Q3: Generalization to Multimodal/Longitudinal Data
> >
> > Yes, we hypothesize that this approach is generalizable. The core problem we identify—**narrative bias** (where models rely on token history over visual evidence)—is intrinsic to the autoregressive nature of current VLMs, regardless of modality (CT/MRI).
> >
> > Our solution—breaking the narrative into independent, visually-grounded queries—is a modality-agnostic strategy.
> >
> > ___
> > ### Q4: Radiologist Preference and Interpretability
> >
> > Regarding radiologist preference, literature such as **Buckley et al. [2]** has shown that the recall of structured radiology reports is significantly superior to unstructured reports. Our work aligns with this clinical trend toward structured reporting.
> >
> > Furthermore, we emphasize that **interpretability** is a direct byproduct of our method. Unlike narrative models that generate a "wall of text," our model allows a reviewer to verify findings topic-by-topic (e.g., query "Heart" $\to$ see "Cardiomegaly" $\to$ verify against the specific "Heart" attention map).
> >
> >
> > ___
> > ### Q5: Inference without Ground-Truth Topics
> >
> > We want to emphasize that our model does **not** assume ground-truth **positive disease labels** (e.g., "Pneumonia exists") at inference. It uses only topics.
> >
> > However, to address the valid concern about the model's reliance on a correct *topic list*, we performed a **Robustness Experiment**. At inference time, we **intentionally injected 2 irrelevant (non-ground-truth) topics** into the query list for every report (e.g., asking about "Bones" in a report that only discusses "Lungs").
> >
> > | Method                                                       | BLEU-1   | BLEU-4   | ROUGE-L  | Micro-F1-14 | Macro-F1-14 | Micro-F1-5 | Macro-F1-5 | F1-RadGraph |
> > | ------------------------------------------------------------ | -------- | -------- | -------- | ----------- | ----------- | ---------- | ---------- | ----------- |
> > | LLaVA-Rad                                                    | 16.5     | 3.6      | 15.3     | 42.1        | 23.3        | 48.5       | 36.6       | 19.8        |
> > |`LLaVA-TA (w/ 2 Random Noise Topics) (report)` | **54.2** | **32.2** | **48.6** | 73.7        | 55.0        | 82.4       | 78.9       | 40.9        |
> > | **LLaVA-TA (Clean Inference) (report)**                      | 43.7     | 24.8     | 42.6     | **80.0**    | **62.4**    | **89.0**   | **86.8**   | **41.4**    |
> >
> > The performance drop is minimal (RadGraph F1: 41.4 $\to$ 40.9). This demonstrates that when queried about an irrelevant topic, the model correctly generates negative findings (e.g., "The bones are unremarkable") rather than hallucinating pathology. This proves the model can function robustly even if an upstream "topic selector" introduces noise.

---

> > > ### Author Response · Authors · 2025-11-24
> > > **Response to Reviewer vVhP (Weakness1)**
> > >
> > > ### Weakness1 : Experimental Fairness (Data Simplification vs. Model Improvement)
> > >
> > > **Answer:**
> > >
> > > To verify the source of our performance gains, we conducted a comprehensive ablation study disentangling **dataset cleaning** vs. **model architecture**.
> > >
> > > We compare the original dataset (**OriData**) against our topic-decomposed dataset (**CleanData**), and compare standard narrative generation (**allGTTopic**, the concatenation of all the ground truth topics) against our proposed topic-guided generation (**singleTopic**, one topic at a time).
> > >
> > > The results based on fine-tuning both MLP and LLM are below:
> > >
> > > **Table 1: Ablation Results (Fine-tuning both MLP and LLM)**
> > >
> > > | Method                         | BLEU-1   | BLEU-4   | ROUGE-L  | Micro-F1-14 | Macro-F1-14 | Micro-F1-5 | Macro-F1-5 | F1-RadGraph |
> > > | ------------------------------ | -------- | -------- | -------- | ----------- | ----------- | ---------- | ---------- | ----------- |
> > > | OriData                        | 40.0     | 16.1     | 30.8     | 58.1        | 40.1        | 57.7       | 50.3       | 30.8        |
> > > | CleanData                      | 46.4     | 26.2     | 40.0     | 63.3        | 40.7        | 66.0       | 57.3       | 32.4        |
> > > | OriData + allGTTopic           | 30.4     | 11.2     | 24.5     | 71.4        | 56.8        | 79.6       | 74.4       | 27.2        |
> > > | CleanData + allGTTopic         | 31.3     | 17.4     | 32.8     | 80.9        | 63.7        | 89.3       | 87.2       | 27.8        |
> > > | **CleanData + singleTopic**    | **50.9** | **32.1** | **60.9** | **83.7**    | **72.2**    | **90.0**   | **87.9**   | **46.5**    |
> > > | CleanData + singleTopic + mask | 50.9     | 31.9     | 60.7     | 83.6        | 71.6        | 89.6       | 87.3       | 46.3        |
> > >
> > >
> > >
> > > **Table 2: Ablation Results (Fine-tuning MLP Only)**
> > >
> > > | Method                            | BLEU-1   | BLEU-4   | ROUGE-L  | Micro-F1-14 | Macro-F1-14 | Micro-F1-5 | Macro-F1-5 | F1-RadGraph |
> > > | --------------------------------- | -------- | -------- | -------- | ----------- | ----------- | ---------- | ---------- | ----------- |
> > > | OriData                           | 16.5     | 3.6      | 15.3     | 42.1        | 23.3        | 48.5       | 36.6       | 19.8        |
> > > | CleanData                         | 9.9      | 1.4      | 10.8     | 31.8        | 17.7        | 35.1       | 24.2       | 10.4        |
> > > | OriData + allGTTopic              | 20.9     | 4.2      | 18.2     | 49.8        | 29.9        | 55.8       | 45.9       | 20.9        |
> > > | CleanData + allGTTopic            | 10.6     | 1.5      | 11.9     | 33.1        | 20.0        | 37.6       | 29.6       | 12.8        |
> > > | CleanData + singleTopic           | 2.9      | 0.4      | 5.9      | 27.7        | 20.4        | 27.5       | 22.3       | 5.8         |
> > > | **CleanData + singleTopic +mask** | **46.1** | **27.4** | **57.3** | **82.7**    | **70.0**    | **89.6**   | **87.5**   | **41.4**    |
> > >
> > >
> > >
> > > These experiments provide three key conclusions that confirm the validity of our approach:
> > >
> > > - 1. **Dataset simplification provides only marginal gains:** Comparing Row 1 (`OriData`) and Row 2 (`CleanData`) in Table 1, we see that simply training on the cleaned data improves the RadGraph F1 score by only **1.6 points** (30.8 $\to$ 32.4). While this helps, it does not account for the majority of the performance leap.
> > >
> > > - 2. **Topic-Guided Generation drives the massive performance boost:** Comparing Row 2 (`CleanData`) to Row 5 (`CleanData + singleTopic`), we observe a massive jump in RadGraph F1 from 32.4 to 46.5. This proves that the performance gain is primarily driven by our single-topic query mechanism, not by the simplified data format. The explicit querying forces the model to focus on factual accuracy rather than narrative style.
> > >
> > > - 3. **Anatomical Masks are essential for parameter-efficient tuning:** Table 2 demonstrates the necessity of the anatomical mask. When the LLM is frozen (MLP-only tuning), the model fails to generate accurate reports using the prompt alone (`CleanData + singleTopic`, RadGraph F1 = 5.8). However, adding the mask (`+mask`) restores performance to state-of-the-art levels (**41.4**). This confirms that our **Anatomy-Aware** design is crucial for guiding the visual encoder when the language model capacity is constrained.
> > >
> > > In conclusion, while dataset structure plays a role, **the vast majority of the performance gain** is attributable to our proposed LLaVA-TA architecture and topic-guided generation strategy, answering the fairness concern.

---

> > > > ### Author Response · Authors · 2025-11-24
> > > > **Response to Reviewer vVhP (Weakness2)**
> > > >
> > > > ### Weakness 2: Attention map quantification
> > > >
> > > > We quantified the quality of our attention maps using the **MS-CXR [3]** benchmark (test subset). The MS-CXR dataset, based on the MIMIC-CXR-JPG dataset, is a well-balanced phrase grounding benchmark dataset that contains carefully curated image regions annotated with descriptions of eight radiology findings, as verified by radiologists. The quantification is evaluated based on the 176 image-annotation pairs, the images of which are all in the test subset.
> > > >
> > > > We use the following three metrics:
> > > >
> > > > (1) **Hit Score (Hit)**, which credits the model if the maximum attention point falls within the ground-truth bounding box (BBox);
> > > > (2) **Contrast-to-noise ratio (CNR)**, defined as $CNR = |\mu_A - \mu_{\overline{A}}| / \sqrt{\sigma_A^2 + \sigma_{\overline{A}}^2}$, where $A$ and $\overline{A}$ represent regions inside and outside the BBox of the attention map, respectively;
> > > > (3) **Soft Precision (SP)**, calculated as $SP = (\sum_{(x,y)\in BBox}S(x,y)) / (\sum_{all(x,y)}S(x,y))$, where $S(x,y)$ is the attention map value at the position $(x,y)$. We also report $SP\text{-}k$, where attention maps are binarized by retaining the top $k\%$ of values before calculating precision.
> > > >
> > > > | Method       | Hit(%)$\uparrow$ | CNR(%)$\uparrow$ | SP(%)$\uparrow$ | SP_thresh1(%)$\uparrow$ | SP_thresh3(%)$\uparrow$ | SP_thresh5(%)$\uparrow$ | SP_thresh10(%)$\uparrow$ |
> > > > | ------------ | ---------------- | ---------------- | --------------- | ----------------------- | ----------------------- | ----------------------- | ------------------------ |
> > > > | LLaVA-Rad    | 29.2             | 37.6             | 22.0            | 31.9                    | 30.7                    | 29.3                    | 27.0                     |
> > > > | **LLaVA-TA** | **59.2**         | **43.4**         | **37.5**        | **62.0**                | **55.7**                | **50.3**                | **41.4**                 |
> > > >
> > > > **We have added more visualization examples in Appendix F and the quantification results in the main paper, Section 4.3.**
> > > >
> > > > As shown above, LLaVA-TA achieves a **2x improvement in Hit Score (29.2% $\to$ 59.2%)**, demonstrating that our topic-guided approach aligns model attention much more closely with human radiologist reasoning than narrative-based baselines.
> > > >
> > > > ___
> > > >
> > > > [1] Constantin Marc Seibold, Simon Reiß, M. Saquib Sarfraz, Matthias A. Fink, Victoria Mayer, Jan Sellner, Moon Sung Kim, Klaus H. Maier-Hein, Jens Kleesiek, and Rainer Stiefelhagen. Detailed annotations of chest x-rays via ct projection for report understanding. In 33rd British Machine Vision Conference 2022, BMVC 2022, London, UK, November 21-24, 2022. BMVA Press, 2022. URL https://bmvc2022.mpi-inf.mpg.de/0058.pdf.
> > > >
> > > > [2] Bryan W Buckley, Leslie Daly, Grainne N Allen, and Carole A Ridge. Recall of structured radiology reports is significantly superior to that of unstructured reports. The British journal of radiology, 91(1083):20170670, 2018.
> > > >
> > > > [3] Benedikt Boecking, Naoto Usuyama, Shruthi Bannur, Daniel Coelho de Castro, Anton Schwaighofer, Stephanie Hyland, Maria Teodora Wetscherek, Tristan Naumann, Aditya Nori, Javier Alvarez Valle, et al. Ms-cxr: Making the most of text semantics to improve biomedical vision-language processing.
> > > >
> > > > URL: http://dx.doi.org/10.13026/b90j-vb87, 2022b.

---

### Official Review · Reviewer_VWZ7 · 2025-11-01

**Soundness:** 3
**Presentation:** 3
**Contribution:** 2
**Rating:** 4
**Confidence:** 5

**Summary:**

This paper identifies a key flaw in current radiology report generation models: their narrative flow causes them to rely increasingly on language context rather than visual evidence, leading to hallucinations and factual errors. To address this, the authors propose LLaVA-TA (Topic-guided and Anatomy-aware), which restructures report generation into independent, topic-specific sentences aligned with corresponding anatomical regions. By guiding the model to generate findings per topic instead of following a continuous narrative, LLaVA-TA significantly improves factual accuracy, visual grounding, and interpretability across multiple radiology benchmarks.

**Strengths:**

The paper starts from a very insightful and underexplored motivation: rethinking why current radiology report generation models tend to overproduce normal descriptions and underrepresent abnormalities. Instead of merely improving architectures or datasets, the authors identify a fundamental bias in the narrative flow paradigm, where sequential language modeling causes models to ignore visual cues and rely excessively on linguistic priors.

The proposed topic-guided and anatomy-aware framework is a clear and elegant solution: by decomposing reports into topic-level units and generating findings region by region, the model enforces stronger visual grounding and factual consistency. This design directly addresses the identified failure mode rather than treating it as noise or data imbalance.

The experiments are comprehensive and convincing, showing substantial gains across factuality and grounding metrics on multiple benchmarks. The presentation is also clear and well-structured, with strong empirical evidence and visual explanations supporting the claims.

**Weaknesses:**

While the paper presents a clear and well-motivated framework, there are a few important limitations that should be addressed to strengthen the work.

First, the method design could be made more comprehensive. Unlike prior work such as RGRG, which explicitly classifies each anatomical region as normal, abnormal, or not described, this paper only generates topic-specific sentences without explicitly distinguishing whether a region is normal or contains pathology. Such classification could make the model more interpretable and closer to real radiology reasoning, where identifying both the presence and absence of findings is equally important. Integrating this step could also prevent redundant or missing descriptions during generation.

Second, the evaluation setup raises potential fairness concerns. The proposed approach simplifies the report structure by rewriting reports into short, topic-level factual statements, removing stylistic and narrative variations. This preprocessing makes the output text inherently easier to match with reference reports under metrics like BLEU or RadGraph F1. In contrast, existing baselines such as R2Gen or RGRG are evaluated on the original, noisier report format, where stylistic diversity and longer narratives make the task harder. To ensure fair comparison, the authors should retrain or evaluate prior models on the same topic-decomposed version of the dataset, or at least quantify how much of the performance gain comes from dataset simplification versus model improvements.

Overall, while the paper introduces a meaningful direction and shows strong results, its methodological completeness and experimental fairness could be improved. A more balanced evaluation, both in terms of report structure and the explicit modeling of normal/abnormal reasonin, would make the contribution more convincing.

**Questions:**

Please see the weakness.

---

> ### Author Response · Authors · 2025-11-24
> **Response to Reviewer VWZ7**
>
> We sincerely thank the reviewer for their constructive feedback. We appreciate the recognition of our "clear and well-motivated framework" and "strong results." Below, we address the specific concerns regarding methodological completeness and experimental fairness.
>
> ### Q1: Methodological Completeness (Region Classification, Interpretability, and Redundancy)
>
> **Answer**:
>
> - For the region classification, while we do not have a dedicated classification head, our model performs **implicit classification** through generation. Crucially, the model is **not only prompted with "positive" topics** during either training or inference time. For example, in Fig.1, the topics "disease-consolidation", "disease-pleural effusion", and "disease-pneumothorax" are all absent. So, the model will have to learn the classification itself.
>
> - On Interpretability (Visual Grounding),  we argue that our method significantly enhances interpretability by enforcing **visual grounding**.
>
>   1. Unlike RGRG, which is region-based but can still generate multiple topic sentences for a single region, our method focuses more on topic-based generation.  The topic-report pair provides interpretations naturally.
>
>   2. By generating short, topic-specific sentences, the model is forced to rely on the image features, rather than hallucinating based on the "narrative flow" of previous tokens.
>
>      To validate this, we quantified the quality of our attention maps using the **MS-CXR [1]** benchmark (test subset). The MS-CXR dataset, based on the MIMIC-CXR-JPG dataset, is a well-balanced phrase grounding benchmark dataset that contains carefully curated image regions annotated with descriptions of eight radiology findings, as verified by radiologists. The quantification is evaluated based on the 176 image-annotation pairs, the images of which are all in the test subset.
>
>      We use the following three metrics:
>
>      (1) **Hit Score (Hit)**, which credits the model if the maximum attention point falls within the ground-truth bounding box (BBox);
>      (2) **Contrast-to-noise ratio (CNR)**, defined as $CNR = |\mu_A - \mu_{\overline{A}}| / \sqrt{\sigma_A^2 + \sigma_{\overline{A}}^2}$, where $A$ and $\overline{A}$ represent regions inside and outside the BBox, respectively;
>      (3) **Soft Precision (SP)**, calculated as $SP = (\sum_{(x,y)\in BBox}S(x,y)) / (\sum_{all(x,y)}S(x,y))$. We also report $SP\text{-}k$, where attention maps are binarized by retaining the top $k\%$ of values before calculating precision.
>
>      | Method       | Hit(%)$\uparrow$ | CNR(%)$\uparrow$ | SP(%)$\uparrow$ | SP_thresh1(%)$\uparrow$ | SP_thresh3(%)$\uparrow$ | SP_thresh5(%)$\uparrow$ | SP_thresh10(%)$\uparrow$ |
>      | ------------ | ---------------- | ---------------- | --------------- | ----------------------- | ----------------------- | ----------------------- | ------------------------ |
>      | LLaVA-Rad    | 29.2             | 37.6             | 22.0            | 31.9                    | 30.7                    | 29.3                    | 27.0                     |
>      | **LLaVA-TA** | **59.2**         | **43.4**         | **37.5**        | **62.0**                | **55.7**                | **50.3**                | **41.4**                 |
>
>      **We have added more visualization examples in Appendix F and the quantification results in the main paper, Section 4.3.**
>
>      As shown above, LLaVA-TA achieves a **2x improvement in Hit Score (29.2% $\to$ 59.2%)**, demonstrating that our topic-guided approach aligns model attention much more closely with human radiologist reasoning than narrative-based baselines.
>
> - On Redundant or Missing Descriptions: Our framework minimizes these errors by design.
>
>   1. Preventing Omissions: By iterating through a fixed, comprehensive list of anatomical topics, the model is explicitly queried about every region, reducing the chance of missing a finding. Moreover, the reference sentences are short and precise after topic split, which leaves less space for the model to omit findings.
>   2. Preventing Redundancy: Since the generation is constrained to a specific topic (e.g., "Heart"), the training objective penalizes the generation of irrelevant or repetitive details (e.g., mentioning lung opacity while discussing the heart).

---

> > ### Author Response · Authors · 2025-11-24
> > **Response to Reviewer VWZ7 (Q2)**
> >
> > ### Q2: Experimental Fairness (Data Simplification vs. Model Improvement)
> >
> > **Answer:**
> >
> > This is a critical insight. We agree that verifying the source of our performance gains is essential. To address this, we conducted a comprehensive ablation study to disentangle the benefits of **dataset cleaning** versus **model architecture**.
> >
> > We compare the original dataset (**OriData**) against our topic-decomposed dataset (**CleanData**), and compare standard narrative generation (**allGTTopic**, the concatenation of all the ground truth topics) against our proposed topic-guided generation (**singleTopic**, one topic at a time).
> >
> > Below are the results based on fine-tuning both MLP and LLM:
> >
> > **Table 1**: **Ablation Results (Fine-tuning both MLP and LLM)**
> >
> > | Method                         | BLEU-1   | BLEU-4   | ROUGE-L  | Micro-F1-14 | Macro-F1-14 | Micro-F1-5 | Macro-F1-5 | F1-RadGraph |
> > | ------------------------------ | -------- | -------- | -------- | ----------- | ----------- | ---------- | ---------- | ----------- |
> > | OriData                        | 40.0     | 16.1     | 30.8     | 58.1        | 40.1        | 57.7       | 50.3       | 30.8        |
> > | CleanData                      | 46.4     | 26.2     | 40.0     | 63.3        | 40.7        | 66.0       | 57.3       | 32.4        |
> > | OriData + allGTTopic           | 30.4     | 11.2     | 24.5     | 71.4        | 56.8        | 79.6       | 74.4       | 27.2        |
> > | CleanData + allGTTopic         | 31.3     | 17.4     | 32.8     | 80.9        | 63.7        | 89.3       | 87.2       | 27.8        |
> > | **CleanData + singleTopic**    | **50.9** | **32.1** | **60.9** | **83.7**    | **72.2**    | **90.0**   | **87.9**   | **46.5**    |
> > | CleanData + singleTopic + mask | 50.9     | 31.9     | 60.7     | 83.6        | 71.6        | 89.6       | 87.3       | 46.3        |
> >
> >
> >
> > **Table 2: Ablation Results (Fine-tuning MLP Only)**
> >
> > | Method                            | BLEU-1   | BLEU-4   | ROUGE-L  | Micro-F1-14 | Macro-F1-14 | Micro-F1-5 | Macro-F1-5 | F1-RadGraph |
> > | --------------------------------- | -------- | -------- | -------- | ----------- | ----------- | ---------- | ---------- | ----------- |
> > | OriData                           | 16.5     | 3.6      | 15.3     | 42.1        | 23.3        | 48.5       | 36.6       | 19.8        |
> > | CleanData                         | 9.9      | 1.4      | 10.8     | 31.8        | 17.7        | 35.1       | 24.2       | 10.4        |
> > | OriData + allGTTopic              | 20.9     | 4.2      | 18.2     | 49.8        | 29.9        | 55.8       | 45.9       | 20.9        |
> > | CleanData + allGTTopic            | 10.6     | 1.5      | 11.9     | 33.1        | 20.0        | 37.6       | 29.6       | 12.8        |
> > | CleanData + singleTopic           | 2.9      | 0.4      | 5.9      | 27.7        | 20.4        | 27.5       | 22.3       | 5.8         |
> > | **CleanData + singleTopic +mask** | **46.1** | **27.4** | **57.3** | **82.7**    | **70.0**    | **89.6**   | **87.5**   | **41.4**    |
> >
> >
> >
> > These experiments provide three key conclusions that confirm the validity of our approach:
> >
> > - 1. **Dataset simplification provides only marginal gains:** Comparing Row 1 (`OriData`) and Row 2 (`CleanData`) in Table 1, we see that simply training on the cleaned data improves the RadGraph F1 score by only **1.6 points** (30.8 $\to$ 32.4). While this helps, it does not account for the majority of the performance leap.
> >
> > - 2. **Topic-Guided Generation drives the massive performance boost:** Comparing Row 2 (`CleanData`) to Row 5 (`CleanData + singleTopic`), we observe a massive jump in RadGraph F1 from 32.4 to 46.5. This proves that the performance gain is primarily driven by our single-topic query mechanism, not by the simplified data format. The explicit querying forces the model to focus on factual accuracy rather than narrative style.
> >
> > - 3. **Anatomical Masks are essential for parameter-efficient tuning:** Table 2 demonstrates the necessity of the anatomical mask. When the LLM is frozen (MLP-only tuning), the model fails to generate accurate reports using the prompt alone (`CleanData + singleTopic`, RadGraph F1 = 5.8). However, adding the mask (`+mask`) restores performance to state-of-the-art levels (**41.4**). This confirms that our **Anatomy-Aware** design is crucial for guiding the visual encoder when the language model capacity is constrained.
> >
> > In conclusion, while dataset structure plays a role, **the vast majority of the performance gain** is attributable to our proposed LLaVA-TA architecture and topic-guided generation strategy, answering the fairness concern.
> >
> >
> >
> > ___
> >
> > [1] Benedikt Boecking, Naoto Usuyama, Shruthi Bannur, Daniel Coelho de Castro, Anton Schwaighofer, Stephanie Hyland, Maria Teodora Wetscherek, Tristan Naumann, Aditya Nori, Javier Alvarez Valle, et al. Ms-cxr: Making the most of text semantics to improve biomedical vision-language processing.
> >
> > URL: http://dx.doi.org/10.13026/b90j-vb87, 2022b.

---

> > > ### Comment · Reviewer_VWZ7 · 2025-11-25
> > >
> > > Thanks for your response. I have one more question: which F1-RadGraph score do you use? Entity F1 or relation F1?

---

> > > > ### Author Response · Authors · 2025-11-25
> > > > **Response to Reviewer VWZ7**
> > > >
> > > > **Question:** Which F1-RadGraph score do you use? Entity F1 or relation F1?
> > > >
> > > > **Answer:** Following LLaVA-Rad[1], we are using relation F1-RadGraph.
> > > >
> > > > [1] https://github.com/microsoft/LLaVA-Rad/blob/main/llava/eval/rrg_eval/run.py

---

### Author Response · Authors · 2025-12-03
**Summary of Rebuttal Updates and Key Contributions (Part1)**

We sincerely thank the Area Chair and all reviewers for their constructive feedback and recognition of our work’s motivation and strong results.



**Paper Summary**: This paper identifies and diagnoses a fundamental limitation in current vision-language models for radiology: **"Narrative Bias,"** where models prioritize linguistic coherence over visual fidelity, often leading to clinical hallucinations (e.g., assuming "edema" follows "cardiomegaly" regardless of visual evidence). To address this, we propose **LLaVA-TA**, a framework that shifts the paradigm from sequential narrative generation to **independent, topic-guided visual querying**. By conditioning generation on specific anatomical regions and breaking sentence-to-sentence dependencies, LLaVA-TA enforces strict visual grounding, achieving SOTA clinical accuracy and significantly reducing hallucinations. Our extensive experiments demonstrate that this approach sets a new SOTA on the MIMIC-CXR-JPG dataset in clinical accuracy (F1-RadGraph: 30.8 $\rightarrow$ 46.5, CheXpert Micro-F1-14: 58.1 $\rightarrow$ 83.7) and OOD generalization (F1-RadGraph on the unseen dataset IU-XRay: 19.8 $\rightarrow$ 45.0), proving that disentangling a report's narrative structure to enforce independent, visually-grounded findings is a crucial step toward building more reliable medical AI.

___

**Rebuttal Highlights:** During the rebuttal period, we conducted **new experiments** and added **extensive new analyses** to address concerns regarding robustness, fairness, and theoretical grounding. We believe these updates significantly strengthen the paper’s contribution as a rigorous, state-of-the-art framework (All the updates are marked blue in the revised paper).

___

**1. Robustness Verified via Comprehensive Stress-Tests**

A primary concern was the potential brittleness of our multi-stage pipeline. We have added a new **Sensitivity Analysis** (Appendix G) that proves our model is highly robust to real-world noise.

To demonstrate our model's stability against the variability inherent in real-world medical imaging, we conducted a comprehensive robustness evaluation from three distinct aspects:

- 1) Upstream Visual Noise (Segmentation Quality Analysis, CXAS)

- 2. Upstream Semantic Noise (Sensitivity Analysis via Topic Dropout, DeepSeekV3)

- 3. Out-of-Distribution Zero-shot Generalization (IU-Xray dataset)

**1.1 `Robustness to Visual Noise (Segmentation Quality Analysis, CXAS)`**

To evaluate robustness against errors in the upstream segmentation module, we quantified the quality of all 224,105 segmentation masks based on their area ratios. We classified masks into **High** ($Z \le 1$), **Mid** ($1 < Z < 2$), and **Low** ($Z \ge 2$) quality groups based on their deviation from the population mean, where $Z=\frac{area\_ratio-\mu}{\sigma}$, $\mu$ and $\sigma$ are the mean and standard deviation for the current segmentation mask's organ group.

The results of the three mask-quality groups are shown below (Detailed grouping information can be found in Appendix G).

**Fine-tuning MLP+LLM**

| Group           | BLEU-1 | BLEU-4 | ROUGE-L | Micro-F1-14 | Macro-F1-14 | Micro-F1-5 | Macro-F1-5 | F1-RadGraph |
| --------------- | ------ | ------ | ------- | ----------- | ----------- | ---------- | ---------- | ----------- |
| LLaVA-Rad       | 40.0   | 16.1   | 30.8    | 58.1        | 40.1        | 57.7       | 50.3       | 30.8        |
| LLaVA-TA (low)  | 52.1   | 32.5   | 60.3    | 80.1        | 56.8        | 91.5       | 86.8       | 43.0        |
| LLaVA-TA (mid)  | 53.5   | 34.9   | 63.2    | 82.9        | 65.5        | 89.4       | 87.2       | 47.9        |
| LLaVA-TA (high) | 50.0   | 30.9   | 59.7    | 84.0        | 71.6        | 89.7       | 87.4       | 45.7        |
| LLaVA-TA (all)  | 50.9   | 31.8   | 60.6    | 83.5        | 71.5        | 89.6       | 87.3       | 44.0        |

**Fine-tuning MLP**

| Group           | BLEU-1 | BLEU-4 | ROUGE-L | Micro-F1-14 | Macro-F1-14 | Micro-F1-5 | Macro-F1-5 | F1-RadGraph |
| --------------- | ------ | ------ | ------- | ----------- | ----------- | ---------- | ---------- | ----------- |
| LLaVA-Rad       | 16.5   | 3.6    | 15.3    | 42.1        | 23.3        | 48.5       | 36.6       | 19.8        |
| LLaVA-TA (low)  | 47.9   | 26.6   | 55.6    | 77.5        | 53.3        | 90.8       | 88.6       | 37.0        |
| LLaVA-TA (mid)  | 49.6   | 31.3   | 60.4    | 82.0        | 62.5        | 90.0       | 88.1       | 44.2        |
| LLaVA-TA (high) | 44.8   | 26.3   | 56.2    | 83.2        | 70.8        | 89.3       | 87.2       | 40.5        |
| LLaVA-TA (all)  | 46.1   | 27.4   | 57.3    | 82.7        | 70.0        | 89.6       | 87.5       | 41.4        |

---

> ### Author Response · Authors · 2025-12-03
> **Summary of Rebuttal Updates and Key Contributions (Part2)**
>
> The above results demonstrate two key findings:
>
> **Correlation with Clinical Accuracy:** Crucially, we observe a positive correlation between mask quality and clinical classification performance (**Micro-F1-14**). In the MLP+LLM setting, the High-Quality group achieves **84.0**, outperforming the Low-Quality group (**80.1**). This validates the effectiveness of our **Anatomy-Aware** design, demonstrating that accurate anatomical grounding directly contributes to diagnostic precision.
>
> **Robustness via Soft Guidance:** Even on the "Low Quality" group (masks with extreme area deviations), the model achieves a RadGraph F1 of **37.0** (MLP setting), which is nearly **2x the performance of the LLaVA-Rad baseline (19.8)**. This confirms our **Soft Guidance** hypothesis: because the segmentation acts as a spatial prompt rather than a hard crop, the model retains access to the global context and remains effective even when masks are imperfect.
>
> **1.2 `Robustness to Semantic Noise (Sensitivity Analysis via Topic Dropout, DeepSeekV3)`**
>
> We found the primary error mode of the pre-processor was **omission**. To stress-test our model, we conducted a **Sensitivity Analysis via Topic Dropout**. We intentionally corrupted the training data by **randomly dropping 1-2 topics from 5% of the training samples**, simulating a high error rate.
>
> | Method                | BLEU-1   | BLEU-4   | ROUGE-L  | Micro-F1-14 | Macro-F1-14 | Micro-F1-5 | Macro-F1-5 | F1-RadGraph |
> | --------------------- | -------- | -------- | -------- | ----------- | ----------- | ---------- | ---------- | ----------- |
> | LLaVA-Rad             | 16.5     | 3.6      | 15.3     | 42.1        | 23.3        | 48.5       | 36.6       | 19.8        |
> | `LLaVA-TA+randomDrop` | 45.4     | 27.0     | 57.1     | 82.7        | 69.7        | 89.5       | 87.3       | 40.9        |
> | **LLaVA-TA**          | **46.1** | **27.4** | **57.3** | **82.7**    | **70.0**    | **89.6**   | **87.5**   | **41.4**    |
>
> The performance drop is minimal (RadGraph F1: 41.4 $\to$ 40.9). This demonstrates that when queried about an irrelevant topic, the model correctly generates negative findings (E.g., "The bones are unremarkable") rather than hallucinating pathology. This proves the model can function robustly even if an upstream "topic selector" introduces noise.
>
> **1.3 `Zero-shot out-of-distribution testing`**
>
> As shown in the main paper (Tables 3 and 4), we test our model on the out-of-distribution IU-X-ray dataset under a zero-shot setting.
>
> LLaVA-TA shows significant improvements on both linguistic metrics (E.g., ROUGE-L from 22.2 to 57.7) and clinical metrics (E.g., RadGraph-F1 from 19.8 to 45.0).
>
> The model generalizes well to unseen image distributions, confirming it is not overfit to MIMIC-CXR artifacts, implying the strong visual grounding ability of the proposed model and the robustness to medical image variations.

---

> ### Author Response · Authors · 2025-12-03
> **Summary of Rebuttal Updates and Key Contributions (Part3)**
>
> **2. Experimental Fairness & Source of Gains**
>
> To verify the source of our performance gains, we conducted a comprehensive ablation study disentangling **dataset cleaning** vs. **model architecture**.
>
> We compare the original dataset (**OriData**) against our topic-decomposed dataset (**CleanData**), and compare standard narrative generation (**allGTTopic**, the concatenation of all the ground truth topics) against our proposed topic-guided generation (**singleTopic**, one topic at a time).
>
> The results based on fine-tuning both MLP and LLM are below:
>
> **Table 1: Ablation Results (Fine-tuning both MLP and LLM)**
>
> | Method                         | BLEU-1   | BLEU-4   | ROUGE-L  | Micro-F1-14 | Macro-F1-14 | Micro-F1-5 | Macro-F1-5 | F1-RadGraph |
> | ------------------------------ | -------- | -------- | -------- | ----------- | ----------- | ---------- | ---------- | ----------- |
> | OriData                        | 40.0     | 16.1     | 30.8     | 58.1        | 40.1        | 57.7       | 50.3       | 30.8        |
> | CleanData                      | 46.4     | 26.2     | 40.0     | 63.3        | 40.7        | 66.0       | 57.3       | 32.4        |
> | OriData + allGTTopic           | 30.4     | 11.2     | 24.5     | 71.4        | 56.8        | 79.6       | 74.4       | 27.2        |
> | CleanData + allGTTopic         | 31.3     | 17.4     | 32.8     | 80.9        | 63.7        | 89.3       | 87.2       | 27.8        |
> | **CleanData + singleTopic**    | **50.9** | **32.1** | **60.9** | **83.7**    | **72.2**    | **90.0**   | **87.9**   | **46.5**    |
> | CleanData + singleTopic + mask | 50.9     | 31.9     | 60.7     | 83.6        | 71.6        | 89.6       | 87.3       | 46.3        |
>
>
>
> **Table 2: Ablation Results (Fine-tuning MLP Only)**
>
> | Method                            | BLEU-1   | BLEU-4   | ROUGE-L  | Micro-F1-14 | Macro-F1-14 | Micro-F1-5 | Macro-F1-5 | F1-RadGraph |
> | --------------------------------- | -------- | -------- | -------- | ----------- | ----------- | ---------- | ---------- | ----------- |
> | OriData                           | 16.5     | 3.6      | 15.3     | 42.1        | 23.3        | 48.5       | 36.6       | 19.8        |
> | CleanData                         | 9.9      | 1.4      | 10.8     | 31.8        | 17.7        | 35.1       | 24.2       | 10.4        |
> | OriData + allGTTopic              | 20.9     | 4.2      | 18.2     | 49.8        | 29.9        | 55.8       | 45.9       | 20.9        |
> | CleanData + allGTTopic            | 10.6     | 1.5      | 11.9     | 33.1        | 20.0        | 37.6       | 29.6       | 12.8        |
> | CleanData + singleTopic           | 2.9      | 0.4      | 5.9      | 27.7        | 20.4        | 27.5       | 22.3       | 5.8         |
> | **CleanData + singleTopic +mask** | **46.1** | **27.4** | **57.3** | **82.7**    | **70.0**    | **89.6**   | **87.5**   | **41.4**    |
>
>
>
> These experiments provide three key conclusions that confirm the validity of our approach:
>
> - 1. **Dataset simplification provides only marginal gains: ** Comparing Row 1 (`OriData`) and Row 2 (`CleanData`) in Table 1, we see that simply training on the cleaned data improves the RadGraph F1 score by only **1.6 points** (30.8 $\to$ 32.4). While this helps, it does not account for the majority of the performance leap.
>
> - 2. **Topic-Guided Generation drives the massive performance boost:** Comparing Row 2 (`CleanData`) to Row 5 (`CleanData + singleTopic`), we observe a massive jump in RadGraph F1 from 32.4 to 46.5. This proves that the performance gain is primarily driven by our single-topic query mechanism, not by the simplified data format. The explicit querying forces the model to focus on factual accuracy rather than narrative style.
>
> - 3. **Anatomical Masks are essential for parameter-efficient tuning:** Table 2 demonstrates the necessity of the anatomical mask. When the LLM is frozen (MLP-only tuning), the model fails to generate accurate reports using the prompt alone (`CleanData + singleTopic`, RadGraph F1 = 5.8). However, adding the mask (`+mask`) restores performance to state-of-the-art levels (**41.4**). This confirms that our **Anatomy-Aware** design is crucial for guiding the visual encoder when the language model capacity is constrained.
>
> In conclusion, while dataset structure plays a role, **the vast majority of the performance gain** is attributable to our proposed LLaVA-TA architecture and topic-guided generation strategy, answering the fairness concern.

---

> ### Author Response · Authors · 2025-12-03
> **Summary of Rebuttal Updates and Key Contributions (Part4)**
>
> **3. Visual grounding quantification**
>
> We quantified the quality of our attention maps using the **MS-CXR* benchmark (test subset). The MS-CXR dataset, based on the MIMIC-CXR-JPG dataset, is a well-balanced phrase grounding benchmark dataset that contains carefully curated image regions annotated with descriptions of eight radiology findings, as verified by radiologists. The quantification is evaluated based on the 176 image-annotation pairs, the images of which are all in the test subset.
>
> We use the following three metrics:
>
> (1) **Hit Score (Hit)**, which credits the model if the maximum attention point falls within the ground-truth bounding box (BBox);
> (2) **Contrast-to-noise ratio (CNR)**, defined as $CNR = |\mu_A - \mu_{\overline{A}}| / \sqrt{\sigma_A^2 + \sigma_{\overline{A}}^2}$, where $A$ and $\overline{A}$ represent regions inside and outside the BBox of the attention map, respectively;
> (3) **Soft Precision (SP)**, calculated as $SP = (\sum_{(x,y)\in BBox}S(x,y)) / (\sum_{all(x,y)}S(x,y))$, where $S(x,y)$ is the attention map value at the position $(x,y)$. We also report $SP\text{-}k$, where attention maps are binarized by retaining the top $k\%$ of values before calculating precision.
>
> | Method       | Hit(%)$\uparrow$ | CNR(%)$\uparrow$ | SP(%)$\uparrow$ | SP_thresh1(%)$\uparrow$ | SP_thresh3(%)$\uparrow$ | SP_thresh5(%)$\uparrow$ | SP_thresh10(%)$\uparrow$ |
> | ------------ | ---------------- | ---------------- | --------------- | ----------------------- | ----------------------- | ----------------------- | ------------------------ |
> | LLaVA-Rad    | 29.2             | 37.6             | 22.0            | 31.9                    | 30.7                    | 29.3                    | 27.0                     |
> | **LLaVA-TA** | **59.2**         | **43.4**         | **37.5**        | **62.0**                | **55.7**                | **50.3**                | **41.4**                 |
>
> **We have added more visualization examples in Appendix F and the quantification results in the main paper, Section 4.3.**
>
> As shown above, LLaVA-TA achieves a **2x improvement in Hit Score (29.2% $\to$ 59.2%)**, demonstrating that our topic-guided approach aligns model attention much more closely with human radiologist reasoning than narrative-based baselines.

---

> ### Author Response · Authors · 2025-12-03
> **Summary of Rebuttal Updates and Key Contributions (Part5)**
>
> **4. Theoretical Grounding & Narrative Bias**
>
> We formally defined **"Narrative Bias"** in Section 2 and provided a new "Blank Image" experiment (Figure 1) demonstrating that standard models ignore visual input as text history grows. We also added qualitative examples illustrating how LLaVA-TA eliminates specific narrative hallucinations (e.g., the "Cardiomegaly $\to$ Edema" shortcut).
>
> Narrative Bias is the tendency of an autoregressive model $P(y_t | y_{<t}, I)$ to maximize the likelihood of the next token $y_t$ based primarily on the linguistic probability $P(y_t | y_{<t})$ rather than the visual conditional probability $P(y_t | I)$.
>
> - Examples of Generative Impact
>
> ​	**Example A: The "Co-occurrence" Hallucination (Cardiomegaly $\to$ Edema)**
>
> | Model     | Reference                                                    | Prediction                                                   |
> | --------- | ------------------------------------------------------------ | ------------------------------------------------------------ |
> | LLaVA-Rad | The patient is status post CABG with intact sternotomy wires.  The  hilar and mediastinal contours appear to be stable with evidence of a tortuous aorta. **There is stable mild cardiomegaly.  There is no pleural effusion or pneumothorax.** There appears to be a subtle increase in opacification in the retrocardiac region, superimposed on a stable mild background of interstitial abnormality, best seen on the lateral view. | **Mild cardiomegaly is overall stable compared to the prior exam.  `There is mild pulmonary vascular congestion with mild pulmonary edema.`  There is no evidence of a focal consolidation. There is no pleural effusion or pneumothorax.**  The visualized osseous structures are unremarkable. |
> | LLaVA-TA  | Sternotomy wires are present. Increased opacification is present in the retrocardiac region. Mild interstitial abnormality is present. Hilar and mediastinal contours are stable. A tortuous aorta is present. **Mild cardiomegaly is present. Pleural effusion is absent. Pneumothorax is absent.** | Median sternotomy wires are present. Mediastinal clips are present. Mild interstitial thickening is present. The hilar and mediastinal contours are normal. **Mild cardiomegaly is present. Pleural effusion is absent. Pneumothorax is absent.** |
>
> LLaVA-Rad sees the enlarged heart and falls into a narrative trap. The sequence "Cardiomegaly $\to$ Edema" is a common linguistic pattern. Despite the visual evidence (only interstitial changes), the model hallucinates edema based on the text probability of the previous sentence.
>
>
>
> **Example B: The "Co-occurrence" Hallucination (pleural effusion $\to$ atelectasis)**
>
> | Model     | Reference                                                    | Prediction                                                   |
> | --------- | ------------------------------------------------------------ | ------------------------------------------------------------ |
> | LLaVA-Rad | comparison is made to the prior study from ___. the feeding tube, left ij catheter and endotracheal tube are unchanged in position. there is persistent cardiomegaly. there is unchanged left retrocardiac opacity. there are no signs for overt pulmonary edema. **there is a small right-sided pleural effusion** as well. overall, these findings are stable. | in comparison with the study of ___, the monitoring and support devices remain in place. continued enlargement of the cardiac silhouette with opacification at the left base consistent with **pleural effusion and `compressive atelectasis`**. no definite vascular congestion. |
> | LLaVA-TA  | the feeding tube is present. the left ij catheter is present. the endotracheal tube is present. left retrocardiac opacity is present. pulmonary edema is absent. cardiomegaly is present. **a small right-sided pleural effusion is present**. | an et tube is present. a left internal jugular line is present. an upper enteric drainage tube is present. opacification at the base of the left lung is present. pulmonary edema is absent. moderate cardiomegaly is present. **small right pleural effusion is present.** |
>
> LLaVA-Rad hallucinates the "compressive atelectasis" right after the disease "pleural effusion", as the phrase "pleural effusion and compressive atelectasis" is commonly seen in the training dataset. As a comparison, by split the topics and cut off the relations, LLaVA-TA will query the image for each topic and generate the findings independently.

---

> ### Author Response · Authors · 2025-12-03
> **Summary of Rebuttal Updates and Key Contributions (Part6)**
>
> **5. Semantic Completeness**
>
> To directly quantify how topic decomposition affects semantic completeness, we performed a **Quantitative Audit** on the original reports.
>
> We manually checked 6 cross-topic findings, and the statistics are as follows:
>
> | findings               | # in original report | #in processed report | Recall |
> | ---------------------- | -------------------- | -------------------- | ------ |
> | Hemidiaphragm          | 17028                | 15764                | 92.6%  |
> | Subcutaneous Emphysema | 2048                 | 1925                 | 94.0%  |
> | Pneumobilia            | 11                   | 10                   | 91.9%  |
> | Widened Mediastinum    | 587                  | 572                  | 97.4%  |
> | Cardiopulmonary        | 35                   | 34                   | 97.1%  |
> | Pleuroparenchymal      | 57                   | 56                   | 98.2%  |
>
> The statistics above demonstrate that our processed reports achieve **>92% recall** even for complex, cross-topic findings. We attribute this high retention to our comprehensive **three-level topic design** (Disease, Organ, Special Categories), which captures findings that do not fit neatly into a single organ.
>
> **Note on the Recall Calculation:** To ensure a rigorous evaluation of factual completeness, we excluded speculative findings (e.g., "suggestive of," "cannot exclude") from the reference set for this analysis. The statistics above reflect our model's high sensitivity (**>92%**) to **definite, actionable clinical findings**.
>
> ___
>
> We are confident that these revisions address the reviewers' concerns and establish LLaVA-TA as a robust, interpretable, and state-of-the-art solution for grounded radiology report generation.

---

### Meta-Review · Area_Chair_n9gc · 2025-12-31

**Summary:**

This paper studies radiology report generation and identifies the common approach of optimizing narrative coherence makes the generation models exceedingly focus on linguistic priors and inter-sentence correlations and prevent them from considering the visual evidence and lead to factual inaccuracies. To address this issue, this work proposes a Topic-guided and Anatomy-aware framework that rewrite reports into a set of  topics and require the model to generate findings for each topic upon the input images and its segmented parts. The experimental result shows that the proposed method can improve the quality of generated findings at the topic- and report-level, mitigating the model’s reliance on narrative flow and strengthen visual grounding.


The reviewers identify several notable strengths of the work. In particular, they commend its insightful and underexplored motivation, the identification of a fundamental bias in the narrative flow paradigm that has not been explicitly quantified in prior work, and the proposal of a clear and elegant solution that is conceptually simple yet effective. The study also offers valuable diagnostic insights into a core weakness of existing approaches. Moreover, the experimental results are comprehensive and convincing, and the presentation is clear and well structured.


At the same time, the reviewers raise a number of concerns. These include that the method design could be made more comprehensive; the evaluation setup may raise potential fairness issues; the novelty and technical depth of the proposed approach are somewhat limited, with the contribution leaning more toward data structuring than algorithmic innovation; and the report-level evaluation remains partially artificial, lacking rigorous assessment of interpretability or validation by human experts. The reviewers also note the absence of a more critical discussion of semantic completeness, that the theoretical grounding of the “narrative flow” hypothesis remains largely intuitive, a reliance on external tools and heuristics, a lack of theoretical justification for the assumption of topic independence, the need of citing more recent works and considering more families of models in addition to LLava.

The authors make a concerted effort to address the reviewers’ concerns, and the response effectively resolves most of the issues raised. In particular, it clarifies aspects related to methodological and semantic completeness, diagnostic recall, sensitivity to upstream errors and ontology choices, generalization to multimodal and longitudinal data, radiologist preference and interpretability, dependence on external tools, error propagation, and transferability. The authors also provide a preliminary theoretical explanation of “narrative bias,” justify the topic independence assumption, clarify the changes relative to LLaVA-Rad, and discuss the connection to medical knowledge graphs. Importantly, additional experiments and analyses are conducted to address key concerns regarding robustness, fairness, and theoretical grounding.

Overall, the response is clear, thorough, and convincing, and the included summary of updates is helpful. That said, the fairness of the experimental comparisons with existing methods and the assumption of access to ground-truth disease labels at inference time would benefit from clearer justification. Also, the theoretical explanation could be further strengthened with deeper and more rigorous analysis, and the relationship to structured radiology reports warrants a more explicit and critical discussion.

Taking all factors into consideration, the Area Chair believes that among the three reviewers who initially expressed reservations (scores of 4, 4, and 4), Reviewers VWZ7 and mzcd would likely increase their scores, while Reviewer vVhP may maintain their original score. Reviewer ccY6, who assigned a score of 6, is expected to maintain this assessment.

In summary, while the work could be further strengthened by ensuring the fairness of the experimental comparisons and by deepening the theoretical analysis, it offers clear value by identifying and analyzing a fundamental bias that has not been explicitly examined in the existing literature, and by proposing a sound and effective solution. The study has the potential to stimulate further research on this issue and to benefit the broader field of medical report generation. Therefore, the Area Chair is pleased to recommend acceptance of this paper for the conference.

**Reviewer Concerns:**

The authors make a concerted effort to address the reviewers’ concerns, and the response effectively resolves most of the issues raised. In particular, it clarifies aspects related to methodological and semantic completeness, diagnostic recall, sensitivity to upstream errors and ontology choices, generalization to multimodal and longitudinal data, radiologist preference and interpretability, dependence on external tools, error propagation, and transferability. The authors also provide a preliminary theoretical explanation of “narrative bias,” justify the topic independence assumption, clarify the changes relative to LLaVA-Rad, and discuss the connection to medical knowledge graphs. Importantly, additional experiments and analyses are conducted to address key concerns regarding robustness, fairness, and theoretical grounding.

Overall, the response is clear, thorough, and convincing, and the included summary of updates is helpful. That said, the fairness of the experimental comparisons with existing methods and the assumption of access to ground-truth disease labels at inference time would benefit from clearer justification. Also, the theoretical explanation could be further strengthened with deeper and more rigorous analysis, and the relationship to structured radiology reports warrants a more explicit and critical discussion.

**Reviewer Scores:**

Taking all factors into consideration, the Area Chair believes that among the three reviewers who initially expressed reservations (scores of 4, 4, and 4), Reviewers VWZ7 and mzcd would likely increase their scores, while Reviewer vVhP may maintain their original score. Reviewer ccY6, who assigned a score of 6, is expected to maintain this assessment.

---

### Decision · Program_Chairs · 2026-01-26

Accept (Poster)